# Feeling lucky? Prospective and retrospective cues for sensorimotor confidence

**Marissa E. Fassold**[1]*, **Shannon M. Locke**[2], **Michael S. Landy**[1,3]

**1** Dept. of Psychology, New York University, New York, New York, United States of America, **2** Laboratoire des Systèmes Perceptifs, Département d'Études Cognitives, École Normale Supérieure, PSL University, CNRS, Paris, France, **3** Center for Neural Science, New York University, New York, New York, United States of America

* marissa.fassold@nyu.edu

**Data Availability Statement:** All data and code files are available on OSF database DOI 10.17605/OSF. IO/XRV65 (https://osf.io/xrv65/).

**Funding:** This work was supported by NIH grant EY08266, awarded to MSL. The funders had no

## Abstract

On a daily basis, humans interact with the outside world using judgments of sensorimotor confidence, constantly evaluating our actions for success. We ask, what sensory and motor-execution cues are used in making these judgements and when are they available? Two sources of temporally distinct information are prospective cues, available prior to the action (e.g., knowledge of motor noise and past performance), and retrospective cues specific to the action itself (e.g., proprioceptive measurements). We investigated the use of these two cues in two tasks, a secondary motor-awareness task and a main task in which participants reached toward a visual target with an unseen hand and then made a continuous judgment of confidence about the success of the reach. Confidence was reported by setting the size of a circle centered on the reach-target location, where a larger circle reflects lower confidence. Points were awarded if the confidence circle enclosed the true endpoint, with fewer points returned for larger circles. This incentivized accurate reaches and attentive reporting to maximize the score. We compared three Bayesian-inference models of sensorimotor confidence based on either prospective cues, retrospective cues, or both sources of information to maximize expected gain (i.e., an ideal-performance model). Our findings primarily showed two distinct strategies: participants either performed as ideal observers, using both prospective and retrospective cues to make the confidence judgment, or relied solely on prospective information, ignoring retrospective cues. Thus, participants can make use of retrospective cues, evidenced by the behavior observed in our motor-awareness task, but these cues are not always included in the computation of sensorimotor confidence.

## Author summary

Sensorimotor confidence is a secondary judgment about how successful we feel a motor action was with relation to the goal. To make this judgment we can draw on information available before we execute an action such as our past experiences and knowledge of the environment, as well as after the action including visual feedback and proprioception, a sense of where our body is in space. In this study, we inquired as to how the information available before and after an action is weighted when considering the final feeling of

role in study design, data collection and analysis, decision to publish, or preparation of the manuscript.

**Competing interests:** The authors have declared that no competing interests exist.

sensorimotor confidence. To do so we asked participants to make reaches to visually cued targets in an unseen-hand task, then report their confidence in how successful they were at hitting the target. We measured each participant's reach accuracy and proprioceptive sensitivity in a separate task. Using mathematical models to fit our data we tested if a given participant depended more heavily on prior information or retrospective information when making their confidence judgment. We found that participants with high proprioceptive uncertainty were more likely to focus on prior knowledge while those with a more exact sense of proprioception incorporated information from both time points.

## Introduction

The efficacy of our actions depends on our ability to quickly separate successes from failures, judging when a mistake was made from scarce information. Because differing amounts of feedback are available from the external world, it is not uncommon that these judgments rely in part on our own metacognitive sense of confidence that reflects feelings of success in the absence of direct external feedback. Metacognition has broadly been the topic of psychological research for over a hundred years [1,2], with confidence becoming a commonly recorded metric in the middle of the last century [3,4]. However, it was not until recently that metacognitive judgements of confidence have evolved from being an aid to interpreting psychophysical data into a focus of research [5,6]. Although confidence judgements can be applied to several mental processes, sensorimotor confidence refers to the confidence we have about the outcome of perceptually-guided motor actions. Sensorimotor confidence is specific to the internal judgment of the success of motor actions that are made with a specific goal in mind, and exists at the intersection between sensorimotor control, perceptual confidence, and motor awareness [7]. When the goal of an action is visually directed, a sensorimotor confidence judgment can be made, as opposed to a judgment of pure motor confidence (e.g., being confident in how high you can jump). Both perceptual and sensorimotor confidence evolve during decision-making, and humans have the capacity to make accurate confidence assessments about the same stimuli at different time points [8–10]. For example, disrupting motor signals during decision making can influence perceptual confidence [11]. However the individual contribution of sensory and motor signals to sensorimotor confidence, and the temporal dynamics of these cues, have received far less scrutiny [7,12,13].

Prospective cues for confidence are available prior to initiating action and are based on information that is predictive of the future outcome (Fig 1). Knowledge of expected motor error [14], a prospective cue to confidence, is accessible and used optimally when planning movements [15] and reporting potential perturbations or mismatches in proprioception and visual feedback [16]. Prospective cues are also intrinsically linked to sensorimotor learning, because knowledge of previous errors continuously updates the current knowledge of motor noise [17], and high confidence in previous performance can influence the current confidence for a different task through sequential effects [13,18,19]. Greater behavioral adaptation can be observed when prospective cues to confidence mismatch task outcome, that is, when visual feedback is decoupled from expectations garnered from proprioceptive information [20], demonstrating that feedback about success influences learning and confidence simultaneously. Confidence mediates the amount of surprise experienced when there is a mismatch between predicted outcome and feedback, and those whose confidence correlates well with their performance are faster learners [13]. In sum, prospective cues to sensorimotor confidence depend on pre-action sensory input and the past experience of the observer. For example, when

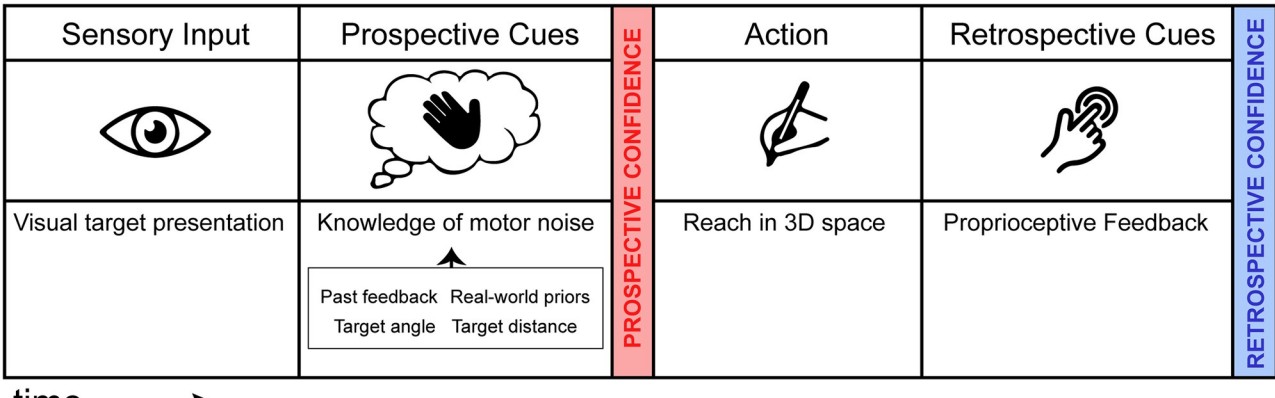

**Fig 1. Temporal order of cue availability for prospective and retrospective confidence.** Prior to an action prospective cues to confidence are available, such as sensory input, and prior knowledge about motor abilities. After the action has been executed, retrospective cues become available including proprioception and sensory feedback when available. Prospective cues can be attributed to any action, while retrospective cues are specific to a single, specific action. Figure created by the authors using a licensed copy of Adobe Illustrator (https://www.adobe.com/legal/terms.html).

determining your confidence in your ability to throw a dart at the bullseye, the distance and size of the board, and your past experience with throwing darts, inform your expected motor noise for that specific toss. These cues, available prior to the action, exist as a set of temporally distinct inputs to confidence.

After an action is performed, a new set of cues become available to the observer (Fig 1). Retrospective inputs to confidence are predominantly used in perceptual tasks where an observer makes a report referring to the accuracy of a just-performed action [2]. These cues are specific to the task just performed, differing from action to action, and include proprioception (the sense of where your body is located in space), visual feedback, and forward models, which are representations of the motor system that use the current state of the system to predict future outcomes [21,22]. These retrospective cues correspond to the most accurate time to query the feeling of confidence because of the higher correlation with performance accuracy compared to prospective assessments [9,23]. An optimal assessment of confidence should combine the prospective and retrospective assessments, thus making use of all available sensory and motor cues from the temporal span of the action.

Confidence is primarily studied retrospectively using a self-report scale following a binary decision task (e.g., responding as to how sure participants were in their statement that a stimulus was present). The multiple sources of noise and multiple degrees of freedom of movement create a complex field of response options and decision outcomes, resulting in layered relations that are difficult to separate out from a simple rating scale. Applying binary decisions to the success of sensorimotor tasks is possible and commonly used (e.g., did the dart hit the bullseye or not?) [7,12,24–26]. However, a more complex but naturalistic approach is also possible by measuring confidence on a continuous scale. For example a dart very close to the bullseye would be considered more successful than a dart that struck the wall.

Traditional models of perceptual confidence are limited by dependance on a binary choice followed by a confidence judgment [6,27]. Use of a binary choice is quite common in perceptual tasks in which a participant categorizes a particular feature of the stimulus (orientation, contrast, presence, etc.), and has also been used for some sensorimotor tasks, where the decision comes after the action as a separate binary choice such as choosing which visually presented trajectory best matched the true motor-performance [e.g., 12]. This approach

introduces natural multidimensional movements but still reduces the choice to a binary decision allowing for correct or incorrect reports. Moving away from binary tasks, another approach that leads to spatiotemporally complex data is to treat the choice of motor plan as the decision [7,15,28]. Treating the motor plan as the decision, the action doesn't have a clear binary correct vs. incorrect plan, and thus the signal-detection-theoretic models of a subsequent confidence response don't apply [7]. In order to incorporate a confidence judgment into a naturalistic reaching task more complex models need to be used. Bayesian approaches to confidence have gained popularity for handling non-binary decisions among other tasks, describing confidence as the posterior probability of being correct given the presence of noise [13,29–35]. Sensorimotor confidence is positioned uniquely to benefit from the Bayesian approach given that measurements in motor tasks are effector specific and recorded in a way that captures the noise naturally present in the system. The sensorimotor literature provides solid evidence for the existence of available priors such as awareness of the body's own kinematics, posture, effect of gravity, possible velocities and motor limits, and strong evidence that participants use priors when making reaches [36–43]; However, less evidence is present for other effectors and priors may be context dependent [44,45]. Each component of a Bayesian model is directly measurable and well researched in the sensorimotor domain. We apply these robust Bayesian metrics to a sensorimotor confidence judgment, giving a novel and interesting perspective on the temporal component of metacognition. Previous research into the components of sensorimotor behavior provides a solid foundation for our extension to a powerful sensorimotor confidence metric.

Assessing the contribution of prospective and retrospective inputs to confidence is best approached using a Bayesian-inference modeling framework. Given the complexity of sensorimotor confidence, the flexibility of the Bayesian-inference framework lends itself to modeling confidence, and given the distinction between pre- and post-performance cues to confidence, it becomes the obvious choice. For sensory-discrimination tasks, Bayesian inference can be used to compute the probability of being correct based on past and present evidence. By analogy, one can combine inputs from multiple sensory sources to compute the expected outcome of a sensorimotor action based on both prior performance (prospective cues) and the likelihood that current performance is better or worse than the expectation (using retrospective cues). This combination results in a metacognitive decision reflecting the internal feeling of confidence. Most studies focusing on sensorimotor confidence, and metacognitive processes in general, use a two-alternative, forced-choice task with a categorical confidence report to operationalize their research question [e.g., 12,46]. The focus on binary decisions made it possible to use confidence directly to measure metacognitive efficiency [6,18,27,47], generate widely applicable dual-decision models [32], and enable connections to animal models [e.g. 48–53] due to the simplicity of the behavioral task [54,55]. However, by pivoting to a continuous measure [e.g., 4,56,57], the data can provide a more nuanced perspective of confidence for sensorimotor behavior and allow for the next steps in developing more behaviorally relevant models [58].

Reaching is an ecologically relevant behavior [59], and confidence judgments are regularly made about reaches in everyday life. However, research is limited on what cues are used to make sensorimotor confidence judgments. Here, we explore the concept that while executing an action, confidence can be assessed at two primary time points: before and after executing the action. We focused on isolating the motor component of sensorimotor confidence and breaking potential inputs into two temporally distinct sets of cues to better understand the contribution of each. We know that humans can easily make confidence judgments about their abilities, and there is substantial interest in the temporal dynamics of metacognition across the field [7,10,32,60–62]. However, the ways in which the cues available at these specific

timepoints (before and after an action) are differently weighed in the final report of confidence have not previously been examined explicitly. We hypothesize that humans make reasonable sensorimotor confidence judgements from monitoring prospective and retrospective cues rather than relying entirely on one or the other, and that sensorimotor confidence judgements accurately reflect objective performance. Reaching is a frequent daily activity, so that humans should have a strong sense of how successful reaches are likely to be, and thus we predict that they will use this prospective information in forming a confidence judgment. Perceptual confidence research has clearly demonstrated an ability to retrospectively evaluate perceptual decisions [6,27], thus we predict retrospective cues will also be incorporated in the confidence judgement. We propose a Bayesian-inference approach to modeling sensorimotor confidence. This is a short leap since Bayesian models of sensorimotor decision-making are readily used and incorporate both perceptual and motor components [16,63]. To extrapolate this to sensorimotor confidence, we need to identify the motor components that are inputs to the confidence calculation [26]. This work will not only advance our understanding of sensorimotor confidence, but also contribute to the broader understanding of the temporal dynamics of metacognition. By integrating Bayesian inference models of perceptual decision-making and confidence with those describing motor decision-making, we have developed a flexible framework to model confidence so that we can focus on the motor inputs and signatures of sensorimotor confidence. Our results show that while almost all participants incorporated prior information when making a confidence judgment, only a subset additionally incorporated trial-dependent measurements to maximize their expected gain. The participants who used this additional information, such as proprioception, had lower noise in the associated measurement than those who did not.

## Material and methods

### Ethics statement

The experimental design and recruitment process of this study were approved by the New York University Committee on Activities Involving Human Subjects. All naïve participants gave informed written consent and were financially compensated per session for their time.

### Participants

16 right-handed participants were recruited from the New York University student population (mean age = 25.5 years, SD = 3 years, two male). All but one participant were naïve to the design of the experiment and only naïve participants are presented as examples in this paper. All participants had normal or corrected-to-normal vision, no physical limitations of the right arm and no self-reported motor abnormalities. All participants were tested in both the control motor-awareness task (Task 1) and the main confidence-judgment task (Task 2).

### Apparatus

Participants sat at a desk in front of a custom-made support frame that suspended a mirror and a frosted glass screen above the surface of the table (Fig 2). Above the screen was a projector (Hitachi CP-X3010) that presented visual stimuli onto the top side of the glass screen, positioned above the mirror. The mirror was equidistant between the screen and the surface of the input tablet device (26.8 X 47.6 cm, Cintiq 22; Wacom, Vancouver, WA) resting on the desk and reflected the underside of the screen (displaying the projection) back toward the observer. This positioning allowed for the stimuli to appear to the observer to be in the same spatial plane as the tablet surface. The participants viewed the stimuli from a head-fixed position with

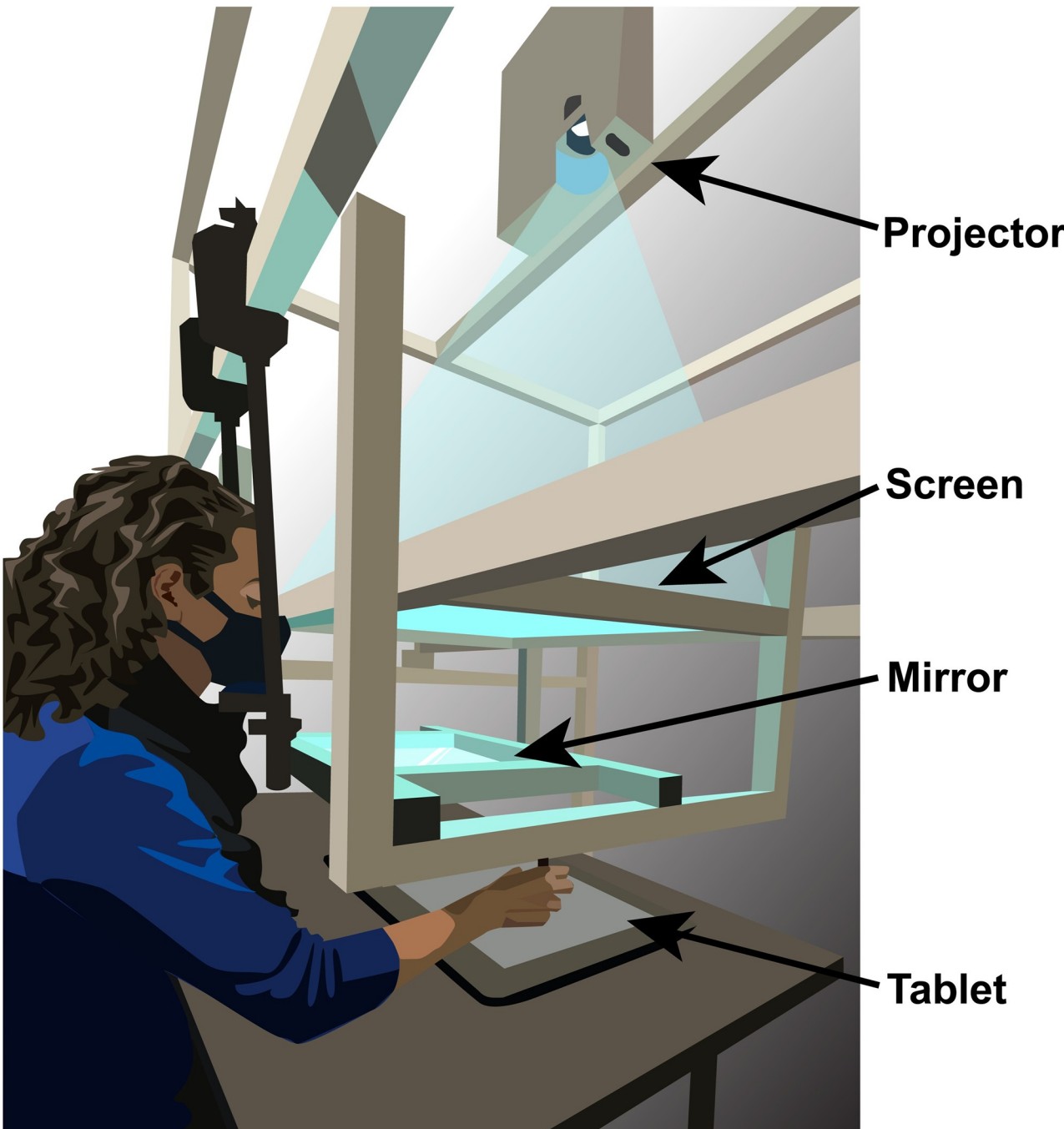

**Fig 2. Apparatus.** Participants responded with reaching movements recorded by a Wacom tablet placed on a desk. A mirror was suspended between a frosted glass screen and the tablet surface in the same horizontal plane. The projector above the glass screen projected stimuli downward. Participants viewed the mirror from a fixed head position and could not see the tablet or their hand while performing reaches with a stylus. The virtual image of the display was in the plane of the tablet. Figure created by the authors using a licensed copy of Adobe Illustrator (https://www.adobe.com/legal/terms.html).

the line of sight above the mirror but below the screen, and could not see their hands during the task. Stimuli were always presented on a mid-gray background. Reach endpoints were recorded with a stylus on the digitizing tablet. The tablet recorded the position of the stylus at a 200 Hz sampling rate. The experimental software was custom written in Matlab (Mathworks,

Natick, MA, USA), using the PsychToolBox extension [64–66] and the WinTabMex toolbox (copyright Kleiner, 2008), and run on the Windows 10 operating system (Microsoft, Redmond, WA, USA, 2015) on a Dell Optiplex 9020 PC. In each session, the tablet was spatially aligned with the on-screen visuals. The calibration was performed by replacing the mirror with a partially transparent half-silvered mirror so that the display and the hand were both visible. Participants used the stylus to touch the tablet at each point in a 3x3 grid of dots across the projected screen area. The resulting data were used to estimate by least-squares an affine transformation between screen and tablet coordinates for each participant. The calibration from each session was only used for data collected during that session to later transform projected stimuli and recorded reach endpoints into the same coordinate frame. Additional responses were made using a Logitech Trackman T-BB18 mouse (Task 1) and a Griffin Power-Mate control knob (Task 2).

## Behavioral tasks

**Control motor-awareness task.** Proprioception has been shown to differ across individuals [67]. The purpose of the control task was to provide an independent measure of the participant's proprioceptive noise. This task queries motor awareness and requires participants to use proprioception to identify a reach endpoint, testing whether observers have access to a proprioceptive retrospective cue. We implemented this by asking participants to report the end-points of unseen reaches and fit a Bayesian-inference model to these reports. The control task was performed during the first session of the experiment and consisted of one block of sixty practice trials where veridical feedback was presented on each trial, followed by five blocks of 60 experimental trials. Proprioception has been shown to improve with repeated exposure to a motor task [67,68], so inclusion of a practice block was necessary. On every test trial, participants placed the stylus inside of a 7 mm annulus centered at the bottom of the screen. Visual feedback of stylus location (a 4.5 mm diameter white dot) was presented when the participant was within 2 cm of the annulus center. After the stylus was in place, a target (a 4.5 mm white dot) appeared 150 mm from the start location annulus, directly in front of the observer. After 1 s, the target turned green, signaling the participant to begin a reach to the target location. During the practice block, participants were presented with veridical visual endpoint feedback after returning to the starting annulus. Feedback was presented for 1 s showing both the target location and the true endpoint of the reach. The experimental trials followed the same format. However, instead of receiving feedback, participants used the mouse to indicate their perceived endpoint location on each trial, guided by visual feedback of the cursor location (Fig 3). The target remained in the same location throughout this task for both the

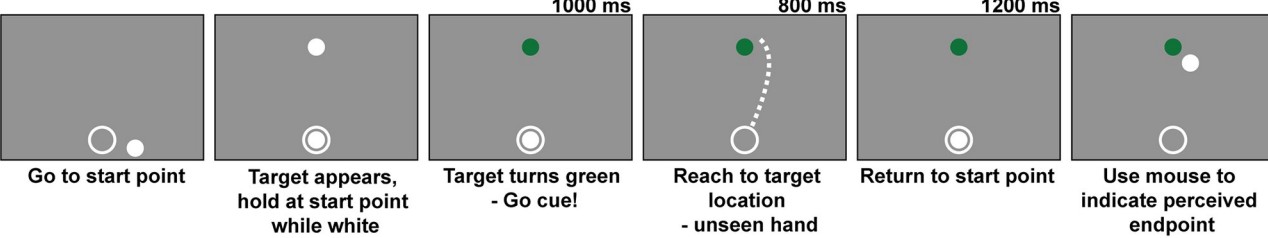

**Fig 3. Control motor-awareness task trial sequence.** Participants moved the stylus into the start point annulus to begin the trial. A target (white dot) appeared straight in front of the starting annulus in the same location on every trial. The target then turned green signaling the participant to begin the reach. The participant then made a reach to the target with the unseen hand, after which they returned to the start point annulus and switched to a mouse to report their perceived endpoint location (white dot) in relation to the target location which was still presented in green. Figure created by the authors using a licensed copy of Adobe Illustrator (https://www.adobe.com/legal/terms.html).

practice and test trials. Trials were rejected and repeated after a visual alert if participants (1) missed the response window (800 ms to start movement from the go-cue and 1200 ms to complete the reach after reach initiation); (2) moved the hand from within the annulus before the go cue; or (3) did not lift their hand off the tablet surface while performing the reach.

**Main confidence-judgment task.** In the main confidence-judgment task, participants reached toward a visually cued target and then they were either shown veridical endpoint feedback or they reported their confidence in how close they landed to the target on a continuous scale. Thus, we have diverged from the typical two-alternative, forced-choice task and instead used continuous error and confidence scales to provide a richer assessment of confidence. We fit a set of Bayesian-inference models to the data to identify which parameters control the cues that inform assessments of confidence. Sessions 2–4 of the experiment were devoted to this confidence-judgment task.

A total of 300 experimental trials were performed during one session, split into five blocks of 60 trials each. A block of 60 practice trials was performed before the start of the experimental data collection to give the observer an opportunity to adapt to the apparatus and familiarize themselves with their own motor error and the task structure. These practice trials were identical to the experimental trials except that in the practice trials the total response window was slowly reduced from 3 s to 2 s to allow participants to get used to the time constraint. On each trial, the participant started with the stylus inside the annulus just as in the control motor-awareness task. A target (a 4.5 mm green dot) was displayed in one of six possible locations on the screen (an arc with a radius of 183 mm and points every 95 mm in 30° increments from 15° to 165° from the starting location at center of the tablet, 42 mm from the bottom edge of the touch screen, where 90° corresponds to straight ahead) with added 2-dimensional Gaussian spatial jitter (SD: 15 mm). The disappearance of the target was the go-cue to reach toward the target location as accurately as possible (Fig 4). The trials were presented in triplets. The first two trials of each triplet received veridical visual endpoint feedback. In the third trial of the triplet, no endpoint feedback was displayed and participants reported their confidence. To make the confidence report the participant adjusted the size of a circle centered on the target location using the control knob (Fig 5). The starting circle size was randomly selected on each trial from a uniform distribution ranging from .5 to 7 cm. Participants were rewarded if the circle intersected or enclosed the 4.5 mm circle that represented the reach endpoint (although this was not displayed on confidence-report trials) and the reward was greater for smaller

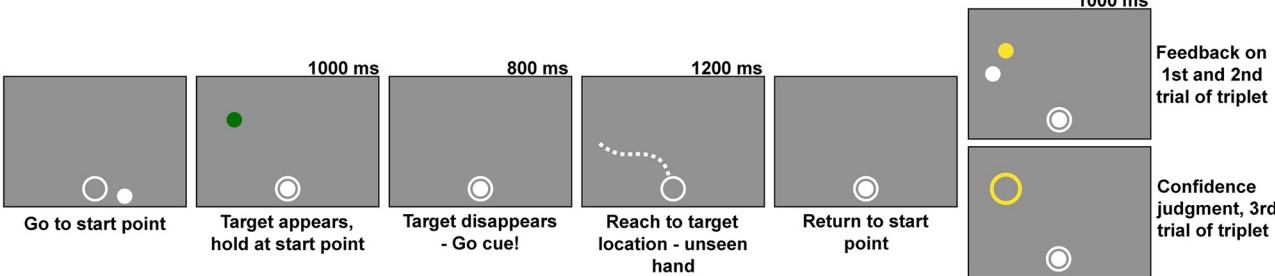

**Fig 4. Main experiment trial sequence.** Participants moved the stylus to a start location annulus at the bottom of the tablet to start the trial. A target (green dot) appeared for 1 s. When it disappeared, participants were required to initiate a reach within 800 ms. After lifting the stylus off of the trackpad and away from the start location annulus, they had 1200 ms to complete the reach (illustrated by dashed line, presented for graphical purposes only as no feedback was given during reach). After reach completion, they returned to the start location annulus and were either shown endpoint feedback (yellow dot) or asked to report their confidence without endpoint feedback (by varying the size of the yellow annulus). The trial was skipped and shuffled back into the remaining possible target locations if the participant left the start location too early, did not start or finish the reach within the allotted time, or did not lift the stylus off the tablet while making the reach. Figure created by the authors using a licensed copy of Adobe Illustrator (https://www.adobe.com/legal/terms.html).

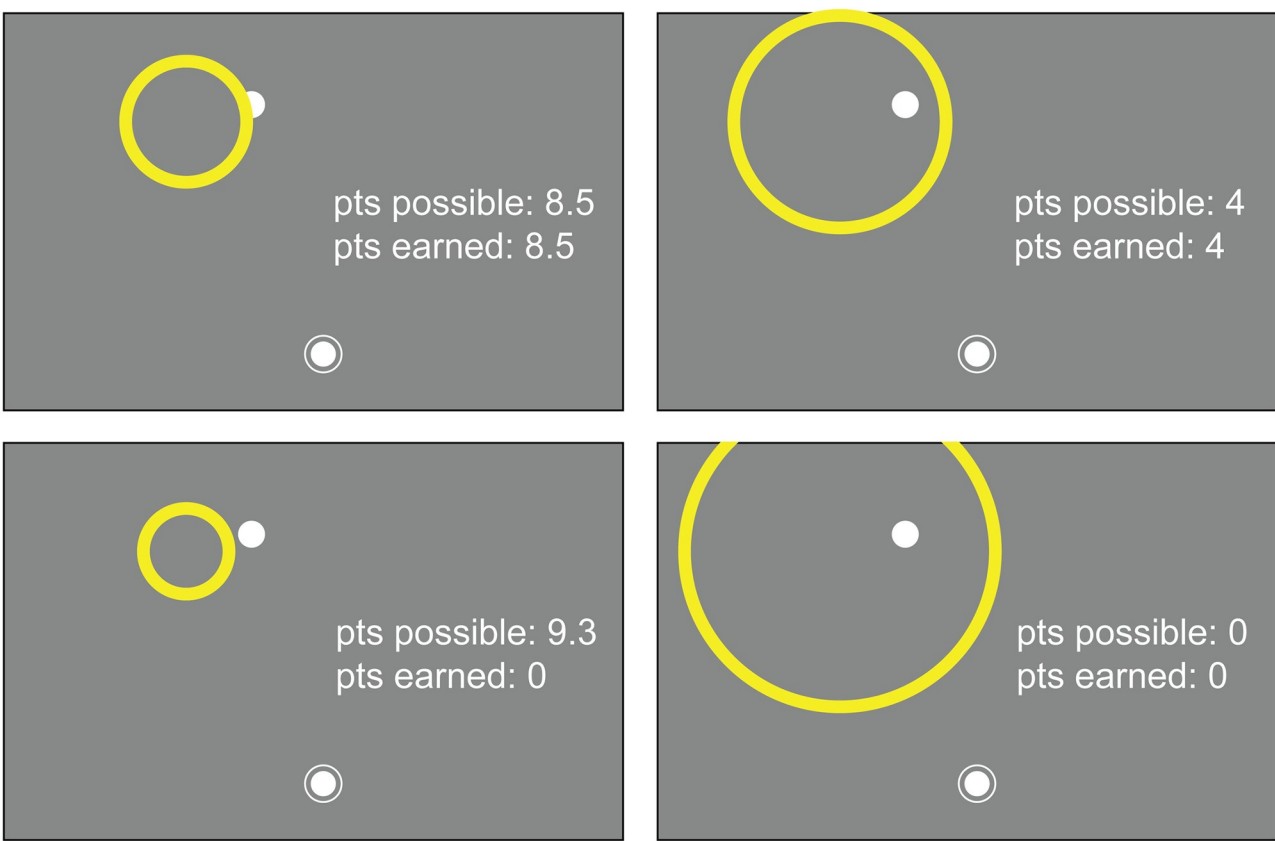

**Fig 5. Confidence report.** Confidence was reported by expanding a circle centered on the target location. Points were determined by the radius, with smaller circles awarding more points (maximum of 10, decreasing linearly with increasing circle size). In a confidence-judgement trial the endpoint circle (shown here in white) was not displayed but is included in this figure for reference purposes. If the confidence circle did not intersect or enclose the endpoint circle, no points were earned on that trial. If the confidence-circle radius was 70 mm or larger, the reward was zero as well. Participants knew the points possible for a given circle size but not the amount of points they had earned thus far. Figure created by the authors using a licensed copy of Adobe Illustrator (https://www.adobe.com/legal/terms.html).

circle sizes. The reward dropped linearly from 10 points for circles with a radius of 2.5 mm or less to 0 points for a radius of 70 mm or more. All six of the possible target sectors occurred in a random order every six trials to ensure an equal number of trials for each sector. The participant was required to return the stylus to the starting annulus prior to receiving feedback or making a judgement about confidence to prevent the use of ongoing proprioceptive signals after the trial had ended. If a participant (1) missed the response window (800 ms to start movement and 1200 ms to complete the reach after reach initiation), (2) moved the hand from the start location before the go-cue, or (3) did not lift their hand off the tablet while performing the reach, the trial was aborted and the target location was shuffled back into future trials to ensure equal testing of all locations. No online visual feedback was presented during the reach.

The relationship between the endpoint, circle size, and points was explained to the participant, and they were instructed to earn as many points as possible given these constraints. This reward design is innovative in sensorimotor research because it centers the point system on accurate performance (greater reward for a small circle size), while still hinging on a reasonable confidence report given the quality of performance in each trial. This made it challenging for participants to game the points system; participants could not purposefully perform badly and then report low confidence to earn points. This point-based incentive promotes

performance accuracy and attentive confidence judgements, a technique that has previously been successful for measuring spatial-memory confidence [69,70] and places the confidence report in the same scale as the error, which allows for direct comparisons between error magnitude and confidence. At the end of each block, the participant's running score was shown, and at the end of the session, a leader board was presented to the participant ranking all previous participants' single-session performance. This was done to motivate the participant to maximize expected gain.

## Performance models

**Control motor-awareness task.** We estimated the participant's motor and proprioceptive noise using the behavioral data collected during the control motor-awareness task. We assumed an ideal performance model that combines prior knowledge of typical reach endpoints and the noisy sensed endpoint location and computes the maximum a posteriori (MAP) estimate of the true endpoint (Fig 6), see Table 1 for summary. The participant aims at the target location, $t$. We assume that the reach is noisy and unbiased and the reach endpoint distribution is isotropic with known motor variance, $\sigma_m^2$. That is, the prior distribution, $p(e)$, of the endpoint, $e$, is $e \sim \mathcal{N}(t, \sigma_m^2)$. After the reach, the participant has a noisy proprioceptive measurement, $p$, of their sensed location, with known proprioceptive variance $\sigma_p^2$. Thus, $p \sim \mathcal{N}(e, \sigma_p^2)$. The corresponding motor and proprioceptive reliabilities are $r_m = \frac{1}{\sigma_m^2}$ and $r_p = \frac{1}{\sigma_p^2}$, respectively. We assume that the variance of the visual estimate of the location of the target is negligible relative to motor and proprioceptive noise and thus treat target location as known precisely. The likelihood function is then $P(p|e) = \phi(p; e, \sigma_p^2)$, where $\phi$ denotes the value of a normal distribution at the sensed location, $p$, with a mean of the true endpoint, $e$, and variance $\sigma_p^2$. The posterior distribution is $P(e|p) \propto P(p|e)P(e)$, and the participant indicates the endpoint position as the MAP estimate, $i$, which is the mean of the posterior:

$$i = \frac{r_p}{r_p + r_m} p + \frac{r_m}{r_p + r_m} t. \tag{1}$$

In this model motor noise, $\sigma_m$, and proprioceptive noise, $\sigma_p$, are the parameters we fit to the data. We estimated these parameters by maximum likelihood. The likelihood of the data, $D$, from a single trial $j$, where $D_j = \{t_j, e_j, i_j\}$, is

$$P(D_j|\sigma_m, \sigma_p) = P(e_j|t_j, \sigma_m)P(i_j|e_j, t_j, \sigma_p, \sigma_m). \tag{2}$$

The likelihood of the indicated endpoint, $i_j$, is:

$$P\left(i_j|e_j, t_j, \sigma_m, \sigma_p\right) = \phi\left(i_j; \frac{r_p}{r_p + r_m} e_j + \frac{r_m}{r_p + r_m} t_j, \left(\frac{r_p}{r_p + r_m}\right)^2 \sigma_p^2\right). \tag{3}$$

Our estimates of a participant's motor and proprioceptive noise are the values of $\sigma_m$ and $\sigma_p$ that maximize the overall likelihood, or equivalently the log likelihood, across all trials, $\{\hat{\sigma}_m, \hat{\sigma}_p\} = \underset{\sigma_m, \sigma_p}{\mathrm{argmax}} \log P(D|\sigma_m, \sigma_p)$. We assume independence of trials so we sum the log likelihood across all trials. Prior to data collection, we simulated the control experiment and determined that 300 trials would be sufficient to accurately estimate these parameter values (see S1, S2 and S3 Figs).

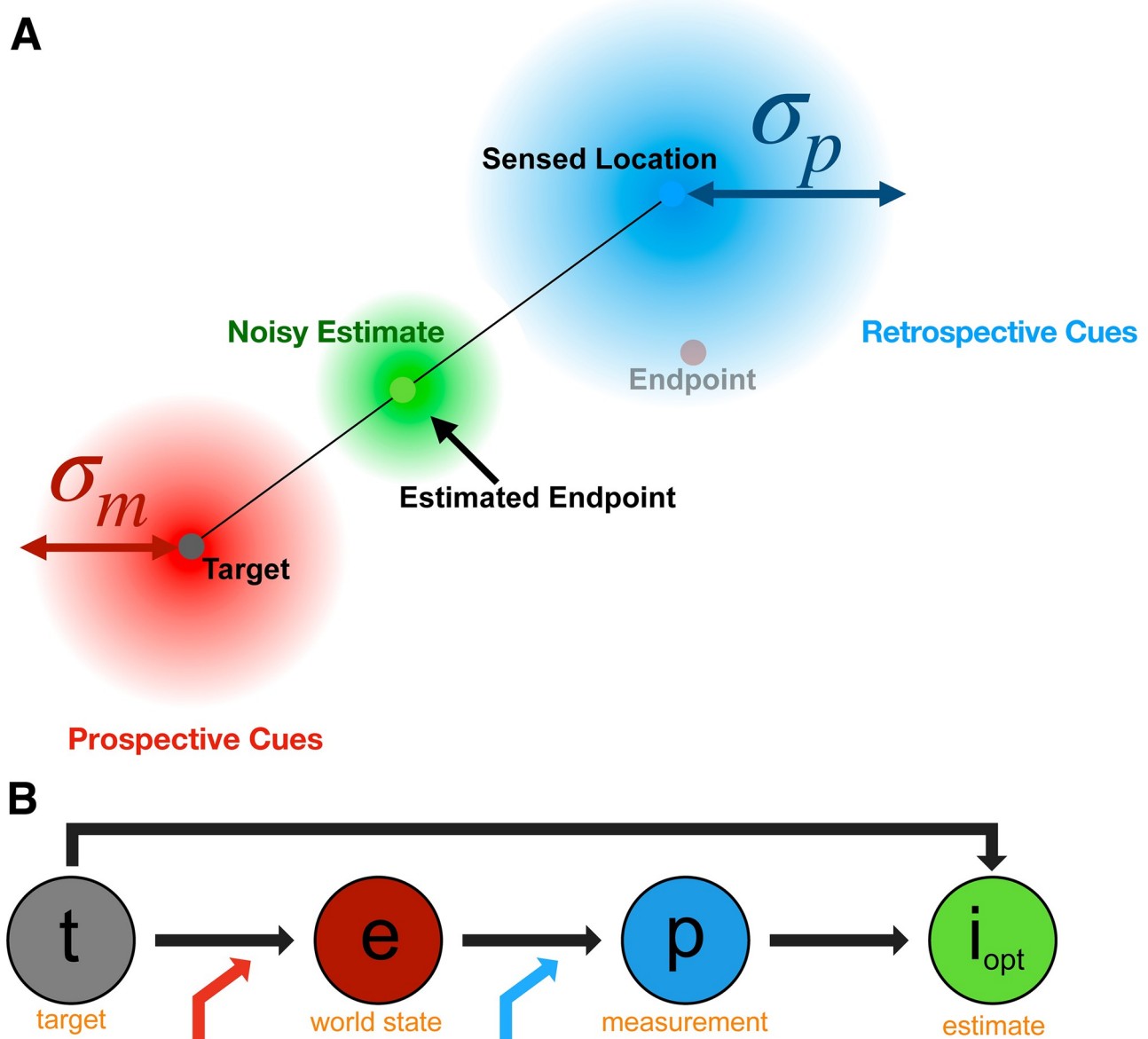

**Fig 6. Cartoon of control motor-awareness task model.** The observer knows the target location and aims for it, resulting in a reach endpoint (red dot) that is perturbed by motor noise (red Gaussian), knowledge of which is a prospective cue along with the target location. The sensed location (blue dot) is where the observer's hand is felt to be located based on proprioception, which has associated variability (blue Gaussian), knowledge of which is a retrospective cue along with the sensed location. Observers combine these two cues to get a noisy estimate of the endpoint (green Gaussian) and indicate the most likely endpoint location (green dot).

**Main confidence-judgment task.** For this task, we assume that the participant aims at the target and then estimates the endpoint based on either retrospective cues, prospective cues, or a combination of both sources of information.

We propose three models to predict the behavioral data (see Table 2 for a summary), all of which compute a circle size that maximizes the expected gain, as a prediction of participant confidence behavior. All models include setting noise, $\sigma_s$, to account for variability in the final

**Table 1. Parameters and equations for the motor-awareness model from the observer's perspective.** This task design forced the observer to use proprioception so this parameter may be isolated during the model fits.

| | Control Motor-Awareness Task Model |
|---|---|
| Inputs | past experience, current evidence |
| Knowledge of $\sigma_m$ | Yes |
| Knowledge of $\sigma_p$ | Yes |
| Prior (observer) | $P(e) = \phi(e; t, \sigma_m^2)$ |
| Likelihood (observer) | $P(\boldsymbol{p}|\boldsymbol{e}) = \phi(\boldsymbol{p}; \boldsymbol{e}, \sigma_p^2)$ |
| Posterior (observer) | $\boldsymbol{i} = \frac{r_p}{r_p+r_m}\boldsymbol{p} + \frac{r_m}{r_p+r_m}\boldsymbol{t};\ \ P(e|\boldsymbol{p}) = \boldsymbol{\phi}\left(\boldsymbol{e}; \boldsymbol{i}, \left(\frac{r_p}{r_p+r_m}\right)^2 \boldsymbol{\sigma}_p^2\right)$ |
| Free parameters | $\sigma_m, \sigma_p$ |

setting of the input knob and assume that the target location is known precisely. The models are:

Ideal Model—assumes that the participant uses both the retrospective and prospective cues available to them to make the confidence judgment (Fig 7).

Retrospective Model—only execution-related retrospective cues are used, giving all weight to the sensed location. This model assumes that the observer has no knowledge of motor error and effectively assumes that it is infinite, i.e., that all landing locations are equally likely (Fig 8).

Prospective Model—only prior prospective information is considered and no trial-specific (e.g., proprioceptive) information is taken into account. This model assumes that the participant can make the confidence judgement prior to beginning the reach based only on knowledge of past experience (Fig 9).

The elements that influence the confidence judgment include true motor noise and its effect on endpoint precision, knowledge of motor noise based on feedback from previous trials, and knowledge of proprioceptive noise. We assume both motor and proprioceptive noise are isotropic, Gaussian and independent of target location. Because participants experience the practice block and receive performance feedback during the experiment, we assume that participants have knowledge of their own motor noise, whether they use it to calculate confidence or not. We use maximum likelihood to estimate model parameters using the data from the control task and the main task together. This resulted in a three-parameter fit for each model because fitting the control experiment requires the motor and proprioceptive noise parameters and all three of the main-experiment models additionally require the setting noise parameter.

**Table 2. Parameters and equations for the Ideal, Retrospective and Prospective models.** Note that the free parameters listed here are those that impact the fit of the data from the confidence-judgment task. Because we simultaneously fit the data from the control task, all models require all three parameters.

| | Ideal Model | Retrospective Model | Prospective Model |
|---|---|---|---|
| Inputs | past experience, current evidence | current evidence only | past experience only |
| Knowledge of $\sigma_m$ | Yes | No | Yes |
| Knowledge of $\sigma_p$ | Yes | Yes | No |
| Prior (observer) | $P(e) = \phi(e; t, \sigma_m^2)$ | $P(e) = constant$ | $P(e) = \phi(e; t, \sigma_m^2)$ |
| Likelihood (observer) | $P(\boldsymbol{p}|\boldsymbol{e}) = \phi(\boldsymbol{p}; \boldsymbol{e}, \sigma_p^2)$ | $P(\boldsymbol{p}|\boldsymbol{e}) = \phi(\boldsymbol{p}; \boldsymbol{e}, \sigma_p^2)$ | $P(\boldsymbol{p}|\boldsymbol{e}) = constant$ |
| Posterior (observer) | $\widehat{\mathbf{e}} = \frac{r_m}{r_p+r_m}\mathbf{t} + \frac{r_p}{r_p+r_m}\mathbf{p}$ $P(\mathbf{e}|\mathbf{p}) = \phi\left(\mathbf{e}; \widehat{\mathbf{e}}, \left(\frac{r_p}{r_p+r_m}\right)^2 \sigma_p^2\right)$ | $P(\boldsymbol{e}|\boldsymbol{p}) = \phi(\boldsymbol{p}; \boldsymbol{e}, \sigma_p^2)$ | $P(\boldsymbol{e}|\boldsymbol{p}) = \phi(\boldsymbol{e}; \boldsymbol{t}, \sigma_m^2)$ |
| Free parameters | $\sigma_m, \sigma_p, \sigma_s$ | $\sigma_p, \sigma_s$ | $\sigma_m, \sigma_s$ |

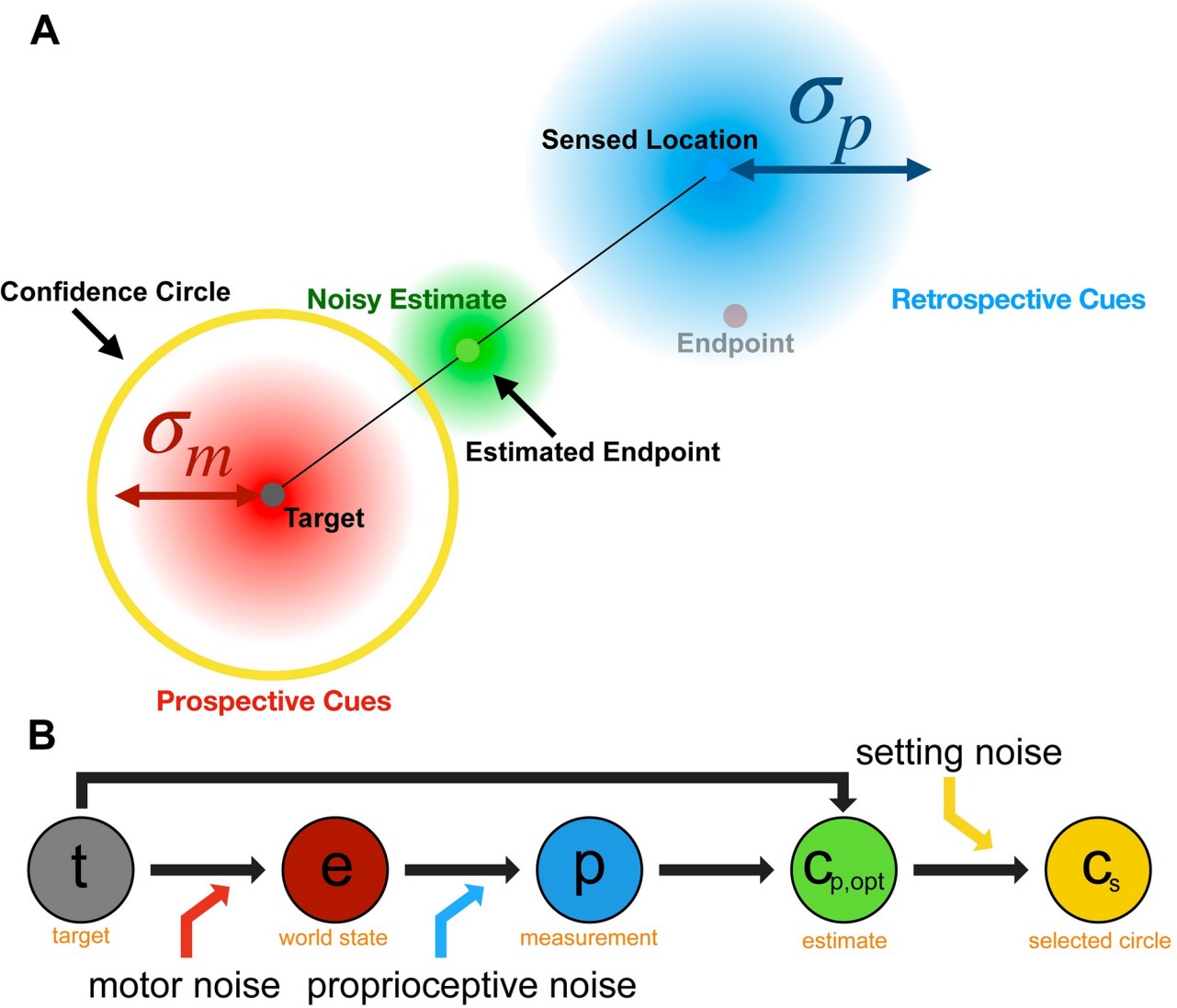

**Fig 7. Cartoon of Ideal performance model from the observer's perspective.** This model assumes that the participant incorporates both prospective (target location and motor noise—red) and retrospective (sensed endpoint location—blue) cues when estimating the endpoint location (green). Figure created by the authors using a licensed copy of Adobe Illustrator (https://www.adobe.com/legal/terms.html).

*Ideal Performance Model.* In the ideal performance model, the participant chooses a circle size that maximizes expected gain using both retrospective and prospective cues. This model's approach earns the most points of the three models.

In this model (Fig 7), the participant knows that reach endpoints are centered on the target with variance $\sigma_m^2$. The MAP estimate of the endpoint $\widehat{e}$ uses the sensed location $\boldsymbol{p}$, the target locations $\boldsymbol{t}$, plus knowledge of motor noise $\sigma_m$ and proprioceptive noise $\sigma_p$. The estimated endpoint is the mean of the posterior distribution:

$$\widehat{\boldsymbol{e}} = \frac{r_m}{r_p + r_m}\boldsymbol{t} + \frac{r_p}{r_p + r_m}\boldsymbol{p}. \tag{4}$$

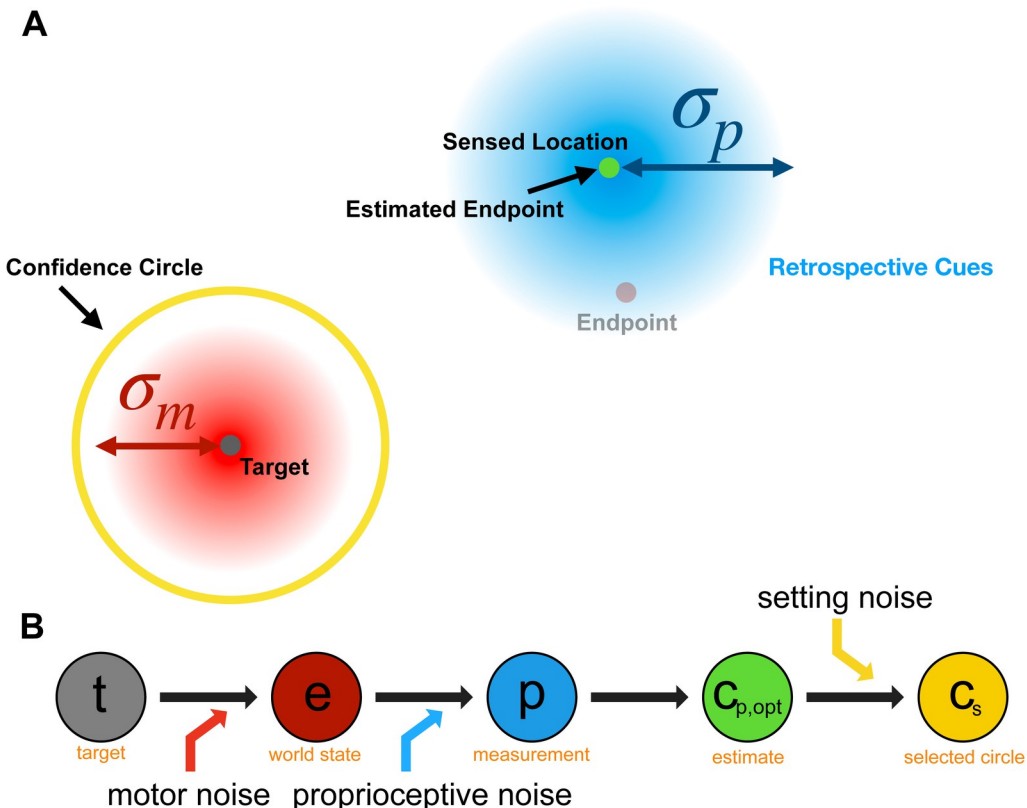

**Fig 8. Cartoon of Retrospective performance model.** This model assumes the decision is only dependent on proprioceptive noise and setting noise, and that motor noise is assumed to be infinite. Consequently, the endpoint estimate is at the sensed location. Figure created by the authors using a licensed copy of Adobe Illustrator (https://www.adobe.com/legal/terms.html).

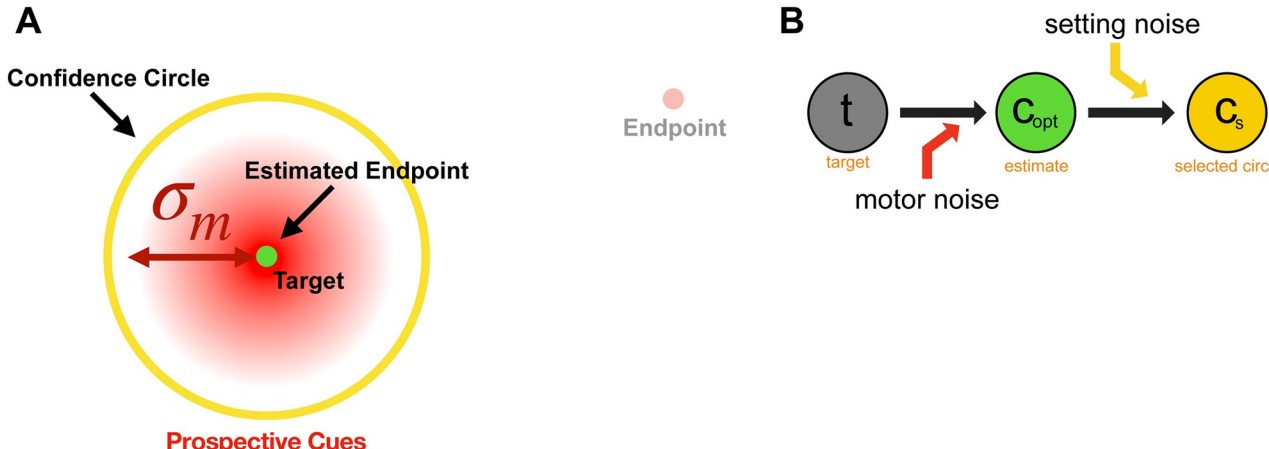

**Fig 9. Cartoon of Prospective performance model.** The participant sets the circle size based only on the known motor noise and does not take any retrospective cues into account as proprioceptive noise is assumed to be infinite. As a result, the intended circle-size setting is constant over trials. Figure created by the authors using a licensed copy of Adobe Illustrator (https://www.adobe.com/legal/terms.html).

The participant's confidence judgment is to pick a circle size $s_c$ so as to maximize expected gain. For our implementation, we note that the optimal circle size only depends on the distance of the sensed endpoint from the target ($\|\boldsymbol{p} - \boldsymbol{t}\|$). Points are awarded on a trial if the circle around the target of size $s_c$, $C(\boldsymbol{t}, s_c)$, encloses or intersects the endpoint circle. The number of points awarded is a function of the circle size, $f(s_c)$. The expected gain for any given circle size is thus

$$EG(s_c|\boldsymbol{p}) = f(s_c) \iint_{C(t,s_c)} p(\boldsymbol{e} = (x, y)|\boldsymbol{p}) dx dy, \tag{5}$$

where $C(\boldsymbol{t}, s_c)$ is the circular locus of endpoints around the target that would lead to reward for circle size $s_c$. The posterior probability of the endpoint is

$$P(\boldsymbol{e}|\boldsymbol{p}) = \phi\left(\boldsymbol{e}; \widehat{\boldsymbol{e}}, \left(\frac{r_p}{r_p + r_m}\right)^2 \sigma_p^2\right). \tag{6}$$

The optimal circle size maximizes expected gain is:

$$c_{s,opt} = \underset{c_s}{\mathrm{argmax}}\, EG(s_c|\boldsymbol{p}). \tag{7}$$

We assume that circle-size settings are perturbed by setting noise, so that the indicated circle size is $c_s \sim \mathcal{N}(c_{s,opt}, \sigma_s^2)$.

*Retrospective Performance Model.* For the retrospective model (Fig 8), endpoints are still generated using $\sigma_m$ however their distribution is not known to the observer. This model assumes a flat prior distribution for endpoint location and hence the perceived endpoint is identical to the proprioceptively sensed location. Thus, for this observer the posterior probability of the endpoint location is centered on the proprioceptively-sensed location:

$$p(\boldsymbol{e}|\boldsymbol{p}) = \phi(\boldsymbol{p}; \boldsymbol{e}, \sigma_p^2). \tag{8}$$

The expected gain for any circle size is calculated using Eq 5, and the optimal circle size again maximizes expected gain (Eq 7), and we again assume that circle-size settings are perturbed by setting noise, so that the indicated circle size is $c_s \sim \mathcal{N}(c_{s,opt}, \sigma_s^2)$.

*Prospective Performance Model.* For this model (Fig 9) the decision is only dependent on motor and setting noise. The model participant does not use the proprioceptive cue, and thus the endpoint posterior probability is

$$p(\boldsymbol{e}|\boldsymbol{p}) = p(\boldsymbol{e}) = \phi(\boldsymbol{e}; \boldsymbol{t}, \sigma_m^2). \tag{9}$$

This likelihood depends only on the target and is independent of the actual endpoint on that trial. As a result, the optimal circle size (Eqs 5 and 7) is constant from trial to trial, and the circle-size setting varies only due to setting noise. This is the simplest model. In this model the observer attempts to set the same circle size on all trials, no matter how far the endpoint is from the target.

**Model fits.** We fit each of our three models to the data from both the control and confidence-judgment tasks and estimated the parameters by maximizing the likelihood of the parameters given the data.

*Ideal Performance Model Fit.* We determined the set of parameters, $\widehat{\boldsymbol{\Theta}}$, that maximized the log-likelihood of our data: $\log P(\mathrm{data}|\widehat{\boldsymbol{\Theta}})$. The parameters were $\boldsymbol{\Theta} = \{\sigma_m, \sigma_p, \sigma_s\}$. The data included the endpoints for each trial, $\boldsymbol{e}_j$, and the selected circle size, $c_{s,j}$. We assume

independence of trials, thus:

$$\log P(\text{data}|\boldsymbol{\Theta}) = \sum_j \log P(\boldsymbol{e}_j|\boldsymbol{t}_j, \sigma_m) + \sum_j \log P(\boldsymbol{i}_j|\boldsymbol{e}_j, \boldsymbol{t}_j, \sigma_m, \sigma_p) + \sum_j \log P(\boldsymbol{e}_j|\boldsymbol{t}_j, \sigma_m)$$
$$+ \sum_j \log P(c_{s,j}|\boldsymbol{e}_j, \boldsymbol{t}_j, \boldsymbol{\Theta}). \tag{10}$$

The first and second terms are summed over the trials in the control task. The third term is summed over all trials in the confidence-judgment task, while the fourth term is summed only over those trials in which a confidence judgment was made. The first and third terms are based on the Gaussian density for each sample endpoint, $P(\boldsymbol{e}_j|\boldsymbol{t}_j, \sigma_m) = \phi(\boldsymbol{e}_j; \boldsymbol{t}_j, \sigma_m^2)$. The second term is defined by Eq 3. The fourth term requires an integration across all possible proprioceptively sensed locations, since the proprioceptively sensed location is unknown to the experimenter. We approximate this integral as a sum over a fine grid of spatial locations:

$$P(c_{s,j}|\boldsymbol{e}_j, \boldsymbol{\Theta}) \approx \sum_{k,l} P\Big(c_{s,j}|c_{s,opt}(\boldsymbol{t}_j, \boldsymbol{p} = (x_k, y_l), \sigma_m, \sigma_p), \sigma_s\Big) P(\boldsymbol{p} = (x_k, y_l)|\boldsymbol{e}_j, \sigma_p)\Delta x \Delta y. \tag{11}$$

For each possible sensed location, an optimal circle size that maximizes expected gain, $c_{s,opt}$, is selected given the distance of the sensed location from the target and the test parameters (Eq 7). The probability of the set circle size, $c_{s,j}$, is computed using the normal density around $c_{s,opt}$ with setting noise, $\sigma_s$, applied. That is, $P(c_{s,j}|c_{s,opt}, \sigma_s) = \phi(c_{s,j}; c_{s,opt}, \sigma_s^2)$. This likelihood is combined with the likelihood of a sensed location appearing at the tested coordinates $P(\boldsymbol{p} = (x_k, y_l)|\boldsymbol{e}_j, \sigma_p) = \phi(\boldsymbol{p} = (x_k, y_l); \boldsymbol{e}_j, \sigma_p^2)$. The log likelihoods from all pairs of $x_k, y_l$ are summed resulting in the log likelihood for a specific set of motor, proprioceptive and setting noise values.

*Retrospective Performance Model Fit.* For this model we also calculate the log likelihood of the test parameters given the sample data (Eq 10). The only difference is that the optimal circle size no longer depends on the motor noise (compare Eqs 6 and 8), but only on the proprioceptively sensed endpoint:

$$P(c_{s,j}|\boldsymbol{e}_j, \boldsymbol{\Theta}) \approx \sum_{k,l} P\Big(c_{s,j}|c_{s,opt}(\boldsymbol{t}_j, \boldsymbol{p} = (x_k, y_l), \sigma_p), \sigma_s\Big) P(\boldsymbol{p} = (x_k, y_l)|\boldsymbol{e}_j, \sigma_p)\Delta x \Delta y. \tag{12}$$

Otherwise, the fit of this model is identical to the ideal performance model.

*Prospective performance model Fit.* For this model we again calculate the log likelihood of the test parameters given the sample data (Eq 10). However, the optimal circle size for this model is a constant value independent of any details of the current trial and depends only on motor noise and thus there is no need to integrate over proprioceptively sensed locations. As a result, the fourth term in Eq 10, $\sum_j \log P(c_{s,j}|\boldsymbol{e}_j, \boldsymbol{t}_j, \boldsymbol{\Theta})$, is replaced by $\sum_j \log P(c_{s,j}|\boldsymbol{\Theta})$, where:

$$\log P(c_{s,j}|\boldsymbol{\Theta}) = \log P(c_{s,j}|c_{s,opt}(\sigma_m), \sigma_s). \tag{13}$$

**Model comparison.** A model comparison was performed using the Bayesian Information Criterion [BIC; 71] on the log likelihood of each data set given the model and best-fit parameters. Since we calculated the maximum likelihood for the control motor-awareness task and main confidence-judgment task data simultaneously, all three models had three parameters and the relative BIC scores are identical to relative log likelihoods. A simulated model recovery was run prior to the data collection showing that all models were identifiable with the number of trials we collected (see S5 Fig).

## Results

### Control motor-awareness task

The goal of the control motor-awareness task was to help constrain estimates of each participant's motor and proprioceptive noise, $\sigma_m$ and $\sigma_p$, since both parameters are needed for modeling confidence judgements in the subsequent task. Participants made repeated reaches toward a fixed target location and reported their perceived endpoint location. We determined the most likely values for each participant using maximum likelihood estimation. The resulting estimates of $\sigma_m$ using the data from the control task alone had a mean of 22.87 mm (SD: 6.17 mm), and the estimates of $\sigma_p$ had a mean of 42.12 mm (SD: 22.76 mm) across the 16 participants. In a pilot study, several participants participated in multiple sessions of the control experiment across multiple testing days and the data indicated that estimates of motor and proprioceptive noise are stable.

### Main confidence-judgment task

Our main experiment investigated how observers use prospective and retrospective cues when making a sensorimotor confidence judgment about a reach. Participants reported their confidence by expanding a circle around the target location until they believed it enclosed their perceived endpoint. We first investigated our participants' performance by comparing their error and confidence circle sizes (Fig 10), in addition to the proportion of total endpoints they successfully enclosed and their efficiency. Average reach error (i.e., distance of reach endpoint from the target) across participants was 20.57 mm (SD: 11.86 mm). The average confidence circle size was 26.93 mm (SD: 7.06 mm). On average, participants captured 75% (SD: 14.7%) of the endpoints within the confidence circle, successfully winning points on those trials.

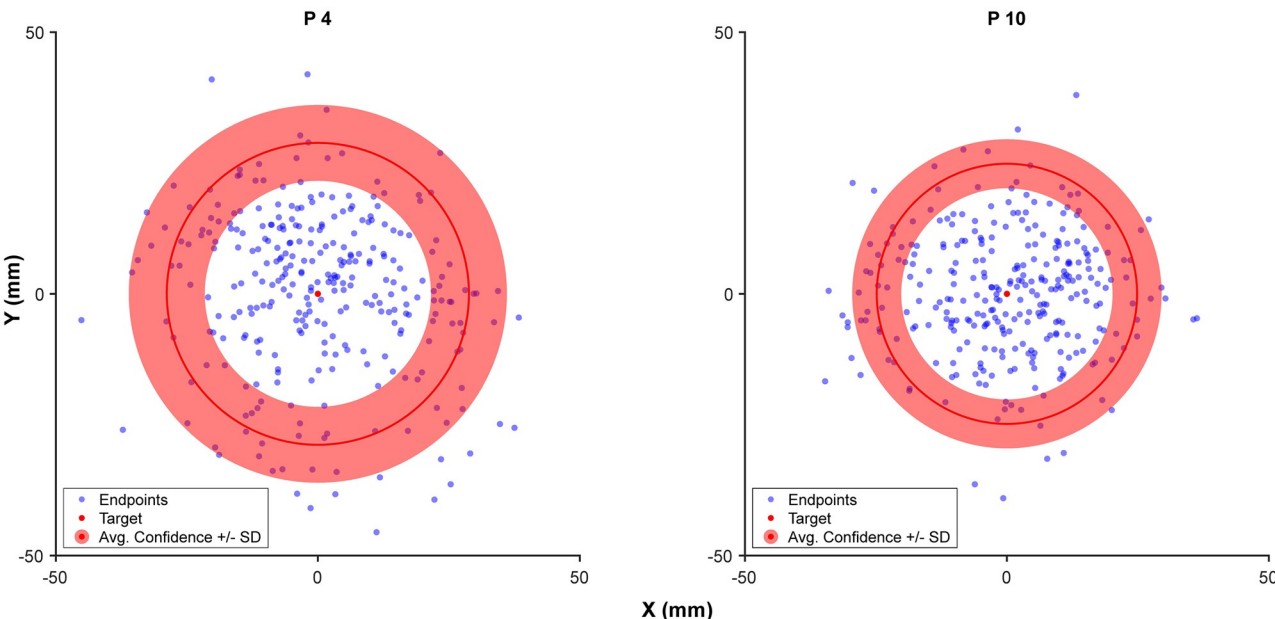

**Fig 10. Scatter plot of reach endpoints relative to the target.** Shown for two sample participants (4 and 10, who will be used throughout the paper) shown in the physical dimensions of the input tablet. The red circle overlay shows the average confidence circle size ± SD for each participant. Participant 4 had a significant correlation between confidence circle size and error, thus reported a wider range of confidence circles which fluctuated with their perceived error across trials. In contrast, participant 10 used a more narrow selection of circle sizes that did not correlate with changes in error across trials.

Participants on average earned 505 points (SD: 107.6). Theoretically perfect performance — capturing the endpoint on every trial with the smallest circle possible — would have resulted in an average of 2171 points (SD: 181), so that participants on average earned 23% (SD: 4%) of possible earned points. Half of the participants showed a significant correlation using the Pearson correlation [72] between confidence reports and error on a trial-by-trial basis ($r^2(298) > .13$, $p < .05$, uncorrected) while the remaining eight showed no significant correlation (Fig 11B).

We assumed that movement endpoint variance was independent of reach direction and isotropic (i.e., a circular Gaussian distribution centered on the endpoint). Our participants have idiosyncratic differences in bias and covariance of reach error, but when rotating all the data to a common reference frame (target direction vs. the orthogonal direction), we found that an unbiased circular Gaussian distribution described participants' behavior adequately (S4 Fig).

## Model fits

Our results are strongly supported by the robust model recovery (S5 Fig), correctly identifying the underlying model for 55 of the 60 simulated data sets (i.e., a 91.7% recovery rate), as well as

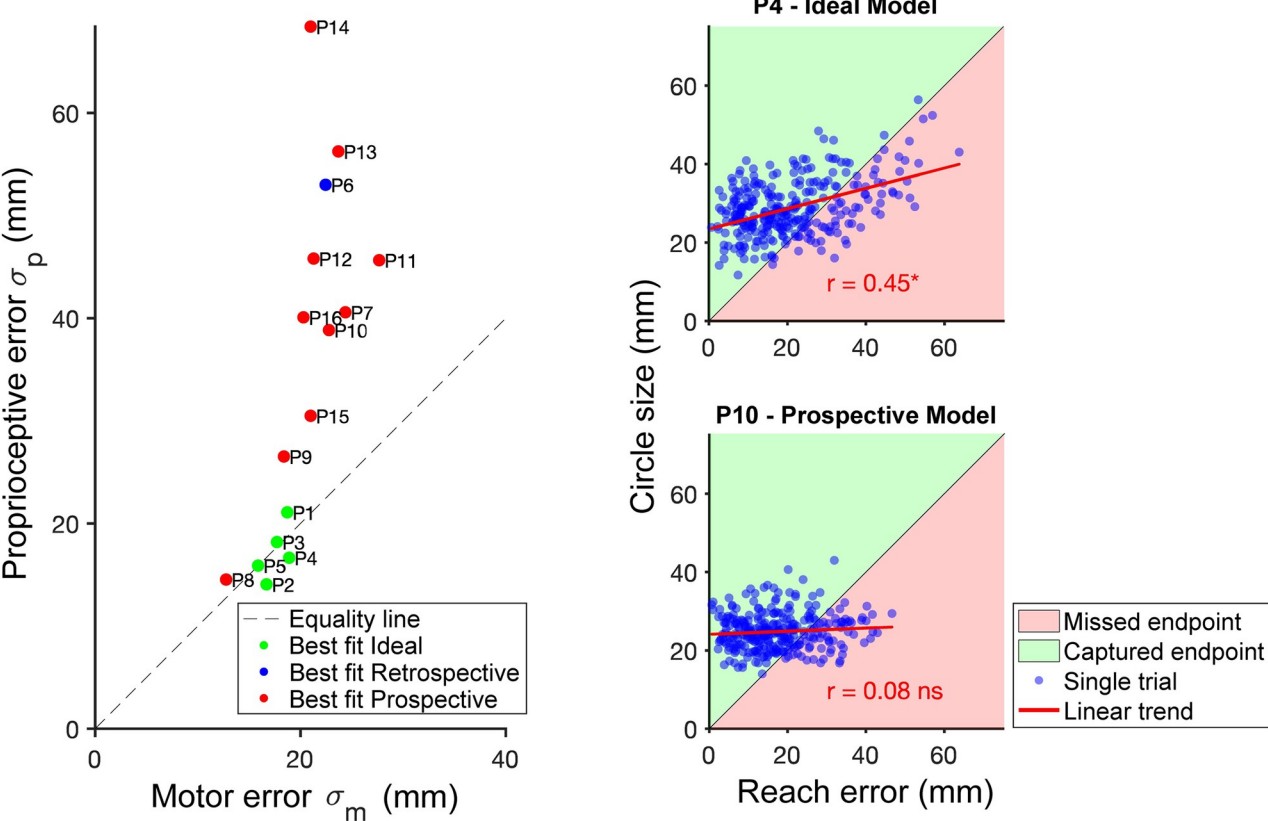

**Fig 11. Scatter plots comparing uncertainty, error and confidence.** (A) Best-fit motor and proprioceptive noise for each participant. Participants best fit by the Ideal model displayed proprioceptive noise close to their motor noise, while those fit best by the Prospective model generally had substantially higher proprioceptive noise. (B) Confidence circle size vs. reach error on each trial for two sample participants in the main confidence-judgment task. P4 was best fit by the Ideal model while P10 was best fit by the Prospective model. The green shaded area reflects judgments where the participant was able to earn points for enclosing their endpoint, and the red shaded area reflects judgments where the confidence circle was smaller than the distance from the endpoint to the target resulting in no points being awarded. Four of five participants best fit by the Ideal model had a significant correlation between confidence and reach error, while only four of the 11 participants best fit by the Prospective model had a significant correlation. (See S7 Fig for other participants.)

the large differences in BIC scores between the models for both the model recovery and the fits of the model to the behavioral data. In the model-recovery simulations, when the Ideal performance model generated the data, there were four data sets that were misidentified. Each of these had a $\sigma_p$ parameter over 60 mm (S6 Fig). Given that zero points are possible when the absolute reach error is over 70 mm from the target, proprioception was uninformative when maximizing expected gain. Such extreme proprioceptive noise values were not at all common in our sample population, so we did not find these misidentifications in our model-recovery analysis to be particularly concerning when we compared models for our behavioral data.

We fit the data from the control and confidence-judgment tasks simultaneously. Thus, one parameter set is estimated for each participant for each of the three models after both tasks were completed. Because we expected individual differences in motor and proprioceptive noise, each participant's data set was fit separately. For each fit we estimated the parameter set yielding maximum likelihood of the data given a model and then computed the resulting BIC scores (Table 3).

In the model comparison there is a clear winning model for every participant. Participants 1–5 were best fit with the Ideal performance model, participant 6 by the Retrospective model and participants 7–16 were best fit by the Prospective model. Participants who were best fit by the ideal model displayed values of proprioceptive uncertainty that were close in value to their motor noise, making them similarly reliable cues. Participants who were best fit by the prospective model usually had proprioceptive uncertainty that was substantially greater than motor noise, resulting in it being a less reliable cue to endpoint location (Fig 11A).

Despite individual differences in reach accuracy, for the most part participants tailored confidence-circle size to their individual motor noise and thus captured the majority of their endpoints. Four of the five participants who were best fit by the ideal model showed a significant correlation between reach error and confidence circle size ($r^2(298) > .13$, $p < .05$), indicating that they based their confidence report on a combination of prospective and retrospective information. Seven of the participants best fit by the prospective model did not show a significant correlation (i.e., $r^2(298) < .12$, $p > .05$). Examples of each are shown in Fig 11B.

For each participant, we simulated performance in the main confidence-judgment task using each of the three models with the best-fitting parameters specific to that participant. Fig 12 shows the resulting distributions of confidence-circle size as well as the corresponding behavioral data for the two sample participants. The winning model fits the data substantially better than the other two models. Additionally we compared various behavioral measures including points earned, proportion of trials where the endpoint was intersected, and average confidence across trials from the participants to those simulated by the models (Fig 13). In general, the best-fitting model predicts the behavioral data better than the other models. To estimate the efficiency of the winning models we simulated task performance for three hypothetical observers. All simulated observers had the same motor variability ($\sigma_m = 20$ mm) and

**Table 3. Model comparison results.** The number in each cell is the BIC score for each model relative to the winning model for that participant. All three models required the same number of parameters. Participants 1–5 were best fit by the ideal model, participant 6 was best fit by the retrospective model and participants 7–16 were best fit by the prospective model. Participants are numbered to group them by the winning model (white background).

| Model | | Participant | | | | | | | | | | | | | | | |
|---|---|---|---|---|---|---|---|---|---|---|---|---|---|---|---|---|---|
| | | 1 | 2 | 3 | 4 | 5 | 6 | 7 | 8 | 9 | 10 | 11 | 12 | 13 | 14 | 15 | 16 |
| | Ideal | 0 | 0 | 0 | 0 | 0 | 45.63 | 93.19 | 219.92 | 380.37 | 550.25 | 257.76 | 679.28 | 221.61 | 173.38 | 343.36 | 148.88 |
| | Retrospective | 372.84 | 487.61 | 411.55 | 426.54 | 498.58 | 0 | 239.33 | 671.99 | 670.36 | 749.18 | 434.63 | 731.11 | 169.27 | 555.87 | 546.51 | 390.12 |
| | Prospective | 323.63 | 190.76 | 396.6 | 290.6 | 315.85 | 62.12 | 0 | 0 | 0 | 0 | 0 | 0 | 0 | 0 | 0 | 0 |

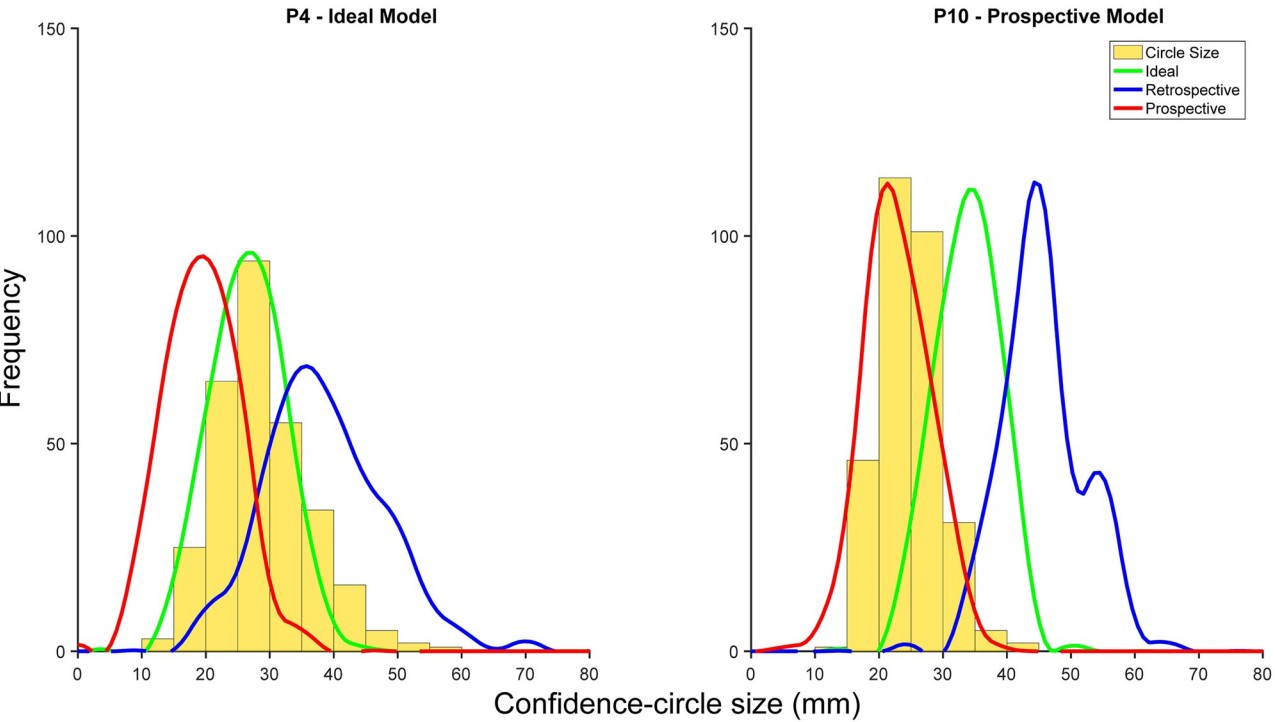

**Fig 12. Data compared to model simulations.** Histograms of selected confidence-circle size (yellow) for two example participants. We simulated behavior for each model with the corresponding set of best-fit parameters for each participant. Curves: resulting distributions of confidence-circle sizes (green: ideal model; blue: retrospective model; red: prospective model). Plots for all participants are provided in S9 Fig. (A) Sample participant that was best fit by the ideal model. (B) Sample participant that was best fit by the prospective model.

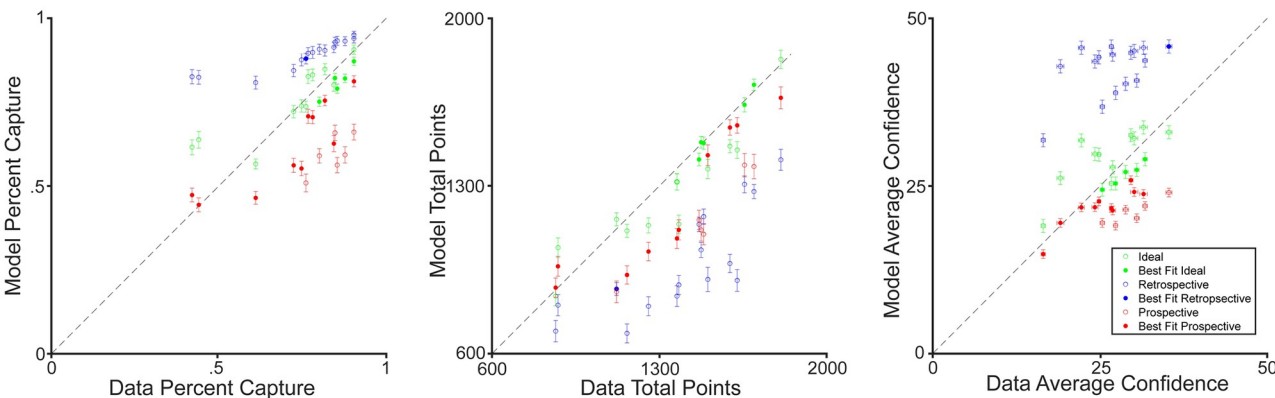

**Fig 13. Capturing behavior with models.** We simulated a total of 9,600 data sets using each participants' best fit parameters, target locations, and endpoints for a given model (200 per participant per model) and compared the model's performance to different behavioral dynamics of the data collected from the participants. In all three plots above there are results from the three simulated models compared to the value measured from each participant, the Ideal Model (green), the Retrospective Model (blue) and the Prospective Model (red). The model that best fit the participant's data in our analysis is denoted with a filled-in circle. (A) The percent of trials in which the confidence circle successfully intersected the true endpoint, thus earning points on that trial. Error bars reflect standard deviation across simulated data sets. (B) The total amount of points earned during the experiment. Higher points are earned for smaller confidence circles, but only if the true endpoint is intersected. Error bars: standard deviation across simulated data sets. (C) Mean confidence across all trials in the experiment. Error bars are +/- SEM across simulated data sets and individual trials.

setting noise ($\sigma_s$ = 5 mm). One observer used only prospective knowledge (the Prospective model). Two observers used retrospective information as well (the Ideal model); but one had low ($\sigma_p$ = 20 mm) and the other had high ($\sigma_p$ = 45 mm) proprioceptive noise. For the low proprioceptive noise observer, ignoring the retrospective information led to an efficiency of 81.6% (2.7 vs. 3.31 points/trial). For the high proprioceptive noise observer, efficiency climbs to 90% (2.7 vs. 3.0 points/trial). Thus, in cases of high proprioceptive noise, the overall performance of an Ideal observer and a Prospective observer is much more similar, as compared to the case where proprioceptive noise is low.

## Discussion

We had 16 participants complete two sensorimotor reaching tasks, first indicating their perceived endpoint (control motor-awareness task) and then making confidence judgments about the success of their unseen reach (main confidence-judgment task). We fit three Bayesian-inference models to the combined data of both tasks that capture how observers use retrospective cues, prospective cues or a combination of the two in assessments of sensorimotor confidence. We found that five of the observers were best fit by the Ideal model, one by the Retrospective model, and 10 were best fit with a Prospective model, indicating the idiosyncratic strategies used by observers.

The results from our model comparison gave credence to our hypothesis that confidence decisions can be based on both prospective and retrospective cues. However we also found that information available at both time points is not always used. By separately modeling the use of prospective and retrospective cues on the determination of the final confidence judgment, we showed that there are important differences in how observers weigh the inputs to sensorimotor confidence at multiple timepoints.

For our model fits of behavioral data, the smallest BIC difference between the winning model and next-best model was 12.38, which is already strong evidence as guidelines suggest a score over 10 may be interpreted as vigorously compelling when the true model is present [73]. The largest of the BIC differences between models for a participant was 749, demonstrating that there are clear-cut patterns in the data discernible by the models. Even with these strong BIC differences we cannot say for certain that any of our models is the true model driving this behavior. While these differences in scores are large there is always the possibility that the true model exists in some other part of model space, and the differences we see here are showing only which model is closer to the underlying true model. That being said, we stand by the reasoning that supports our models and the conclusions we have presented here. Since the BIC was calculated over the data from both tasks (i.e., control and main tasks), all three fits used three parameters ($\sigma_m, \sigma_p, and \, \sigma_s$) so that there was no model penalized for model complexity. Note that participant 6 had the most similar model predictions of the distribution of circle sizes of any of the fits (see S9 Fig for plots of all subjects' model comparisons) with the smallest BIC differences between models. This participant verbally expressed difficulty with the task so it is possible that different strategies were used by this observer across sessions, which resulted in the final overlapping model fits.

The behavioral analyses we performed support the conclusions of our model fits. Specifically, we saw that participants best fit by the ideal performance model more often showed significant correlations between reach error and confidence-circle size, while participants best fit by the prospective model showed this significant correlation less than 50% of the time. This provides further support that the model fits distinguish between participants who use trial-specific retrospective inputs to determine their confidence versus those who only use fixed, prospective information.

People are natural experts at reaching because we practice reaching every day of our lives [59]. Due to the wealth of background experience, it is not surprising that when judgments are made about sensorimotor confidence, prior experience strongly influences the feeling of success. A participant's estimate of their own motor noise likely comes from a lifetime of reaching experience as well as from the feedback of movement endpoint during our tasks. Even though we assume that our feelings of confidence are a byproduct of our actions, we found that many participants gave little weight to their proprioceptive signals in the absence of vision. If we had instead allowed participants to have strong visual feedback, making the task essentially trivial, we would have expected this highly-reliable cue to be used by the observers. However, the importance of proprioception itself should not be overlooked as it can elicit improvements in performance when an action is known to have low accuracy [68,74]. Our Bayesian model builds upon previously well-researched sensorimotor inputs to incorporate both prospective and retrospective information into a final judgment of sensorimotor confidence while using a complex decision in the form of a reach and Bayesian inference. By eliminating visual feedback, we ensured the prospective and retrospective cues were more commensurate in terms of reliability, which allowed us to better identify the prospective and retrospective inputs to sensorimotor confidence.

Previous research suggests that confidence judgments can be biased [27]. Additionally, self-reported confidence judgments often display inter-individual variability [56,75], and most scale-based reports cannot be directly compared to the true amount of error, increasing this variability. By centering our confidence circle on the target location, we could directly compare the size of the circle and the size of the error, allowing direct comparisons across observers. Previous research has shown that incentivizing accurate confidence judgments increases metacognitive sensitivity while concurrently increasing confidence bias [76], where accurate confidence judgments are defined as those correlated with performance. Significant correlations between error and confidence size observed for some observers led to more accurate confidence reports, and in turn higher points. However we do see that even for those participants who showed significant correlations between error and confidence there were noticeable under-predictions of error. For all participants, when error was over 40 mm, circle-size settings were nearly always too small to capture the endpoint. That is, participants would rather take the risk for such large errors of setting a smaller circle size and hoping to occasionally capture the target. This inclination towards risking a possible zero point score in order to potentially earn at least four points shows that while metacognitive sensitivity may have been increased as a result of our using incentives, there was still a bias in the confidence report toward higher-reward circles.

## Performance strategies

The specific time points used to form the final feeling of confidence in the success of an action can differ from person to person, often dictated by the amount of uncertainty they have in their proprioceptive signal, a retrospective cue. Based on the results of our model fits we see that participants whose motor and proprioceptive noise are commensurate are more likely to use both prospective and retrospective information. For such a person, it is beneficial to hold off a confidence decision until retrospective cues become available (i.e., after the action is complete). Because of the validity of their proprioceptive information, this additional retrospective information reduces the uncertainty of an estimated endpoint location when combined with prospective information. Participants who had a very noisy proprioceptive signal tended toward a strategy that depended more heavily on prospective performance cues, ignoring noisy retrospective signals. Additionally, during the confidence report participants' hands had

already moved back to the starting location away from the reach endpoint, so their report is based on the remembered proprioceptive location. This additional step may additionally explain why proprioception was not actively taken into account by all participants.

Having different best-fitting models across participants is a common phenomenon in cue combination and multi-sensory research [e.g., 77,78], as well as perceptual confidence [35,46]. As we have split our motor signals into two distinct cues (prospective and retrospective), it is not surprising that we also see a divergence in strategies reflecting inter-individual differences as seen in cue-combination studies. In sensorimotor research inter-individual differences and variations between best-fitting models can be found in responses to adaptation [79], and across age and skill groups [80,81]. The consideration of the computational cost of including additional information also mirrors that discussed in terms of heuristics decision making [82], and in particular, judgements of confidence. Often, observers are found to deviate from more complex ways of deriving confidence, usually by inferring the difficulty of a particular judgement from stimulus-based cues [e.g., 83–86] or applying approximations to the relationship between noise and decision accuracy [e.g., 33]. Our results suggest the participants may have similarly selected an easier, non-optimal strategy based on idiosyncrasies in the validity of proprioception.

## Confidence and reward strategies

When a participant's proprioceptive noise is high it's possible that these participants had less to gain from incorporating retrospective information because they could still win nearly the same mean point reward with similar circle settings given that their accuracy was consistent across reaches. If it takes extra effort for a participant to account for the retrospective cues, it's possible that those with decent accuracy and less reliable proprioception see it as not worth the effort, while those with more reliable proprioception may benefit more from the extra effort as the additional expected gain is seen as worth the mental cost. To test this theory we looked to our simulation, which demonstrated that it is substantially more costly to ignore retrospective cues when proprioceptive and motor noise are commensurate. Given that there is a reward linked to performance and confidence collectively, which we expect participants to maximize, there are payoffs to be considered by the participant beyond their own expectations of their own skill and confidence alone [46,76,87].

Generally the metric for a good confidence judgment is one that correlates with performance [9], however there can be variability from person to person in how they strategically implement their confidence ratings, especially when reward is involved [46]. In this experiment points gain is most efficient when confidence judgements are well matched with performance, with a penalty for overconfidence (zero points earned when confidence circle radius is smaller than error distance). Overconfidence is prevalent in planned motor tasks [88], and by thus penalizing overconfidence with a reward reduction we emphasized the need for a strategy on behalf of the participant.

If a participant is risk-sensitive in the face of uncertainty they may opt for a strategy to amass an intermediate amount of points on all trials, rather than the maximum possible on each trial, implementing this under-confidence in their performance as a systematic safe bet, because earning some points on every trial could be more appealing than earning a mix of high and null rewards [28,89]. This strategy would decrease the correlation between reach error and circle size. However, additional information becomes available as the trial progresses but if unreliable it can be ignored. Thus, participants may opt for a simpler strategy if additional information is not beneficial, with a similar consequence on the error-confidence correlation. If you are aware of your proprioceptive uncertainty and yet choose to ignore

proprioceptive input, this may not be a sub-optimal approach. The minimal gain in points from using proprioception might be outweighed by the extra effort required to factor in that input. Given what we have shown regarding the differences in proprioceptive sensitivity across our observers and how that correlates with strategy, this supports the idea that participants forego use of proprioception when its value is low compared to the effort required to use it.

## Models

Metacognitive judgements of confidence in perceptual decision making are predominately modeled using a version of signal detection theory (SDT) [90–92] with added parameters allowing for a graded evaluation of decision quality, where signal strength is a determining factor of confidence. SDT does not take into account any online information or segment the time point at which information is incorporated. This highlights an important limitation of this approach as sensorimotor confidence is not necessarily determined fully from end-of-trial values. Accumulation-to-bound models, such as the popular drift-diffusion framework, which extend SDT by incorporating a temporal factor, do take the time into account [50,55,93]. There are even extensions to the standard DDM model to account for nuances in metacognition by allowing accumulation to continue after the decision is reported but before the confidence judgment, allowing the observer to seek further information for the second-level judgment [94–97]. There have been several comparisons of these varied approaches to modeling confidence judgments [33,98,99]. Of these popular models only Bayesian confidence offers the flexibility to be extended to sensorimotor confidence, and could even be extended to capture moment-by-moment changes in metacognition through the use of Kalman filters [100] or similar extensions. Our choice to use a Bayesian-inference approach is well supported by previous work as sensorimotor confidence is a plausible extension of sensorimotor decision-making given that the perceptual and motor components are already incorporated in the framework [16,32,44,63,101]. This prior sensorimotor research considers both the temporal resolution of individual trials and within-trial action dynamics, highlighting the temporal flexibility of the Bayesian approach.

Our Bayesian framework allows for a dynamic measure of metacognition [62,102] by incorporating additional information as the trial progresses. Before an action an observer may have beliefs about the environment and potential outcomes [28]. Since knowledge of motor error is a robust prior that is known to be incorporated into prospective judgments of performance [15] we expect prospective cue use. Further evidence, generated by the action, could then be accumulated throughout the trial, or not, depending on the metacognitive strategy of the observer [e.g., 7,10,60]. The advantage of our model is that while it is complex as a whole, the inputs are very simple and could be applied to a multitude of motor and perceptual decisions including reward under risk, visual judgments with prior probabilities, and confidence in motor awareness.

## Placement within sensorimotor confidence

Sensorimotor confidence is the feeling of success from a motor action with a specific goal in mind, with a rating specifically hinging on self-evaluation of performance [7,86]. An overlapping and commonly conflated field of research is sensorimotor awareness, which pins confidence ratings on the validity of a perceived result (e.g., confidence that the displayed endpoint was their true endpoint, or a displayed reach trajectory matches their action) instead of the success of the goal [e.g., 12]. In this study our participants evaluated their own performance, with their action relating directly to a goal, which puts us solidly in the realm of sensorimotor confidence. Specifically we centered the confidence circle on the target location so a report of

high confidence (small circle) means the participant feels their true endpoint is close to the target, while low confidence rating (large circle) means the participant feels their true endpoint was far from the target. It's important to note that a participant could feel very confident in where their true endpoint was, however if it was far away from the target they would report low confidence in achieving the goal of hitting the target. We also separately perform a motor awareness task to jointly fit motor and proprioceptive error in conjunction with our confidence metric. To further elucidate the difference between awareness and confidence our results indicate that separate portions of perceptual systems may be used by the same observer depending on whether they are calculating awareness or confidence in their reach endpoint. Specifically, while all participants can use proprioception when required by the motor-awareness task, they do not necessarily take it into account when making a confidence judgment. By highlighting the differences between participants in the reliability of their individual priors and measurements we have found that confidence systems may not be a one-size-fits-all paradigm. Our experimental design allows us to keep the richness of the action by not compressing to a binary choice, making us able to evaluate performance as a whole and use dynamic physical measurements. These design choices make judgments of sensorimotor confidence in our task more similar to those experienced in the real world.

## Future directions and real-world applications

It's important to consider that the use of cues in a lab setting might be less fluid than in the outside world where the environs shift rapidly. When playing catch, the wind might pick up, or the sun shine in your eyes, making one sense or another less reliable. If someone is depending heavily on their motor priors but now find themselves in an environment with new rules applying to the physics of movement, would more weight be applied to trial-dependent retrospective cues? Or, if proprioception mismatched performance, would existing priors become more valuable? These questions are important because in the real world motor noise and proprioception are not predictably static. Development, aging, illness, intoxication, injury, and environment all have an affect on our motor and sensory signals, often creating a mismatch between our expectations and reality. How might these changes affect confidence? An assumed motor error lower than one's true motor error (such as resulting from intoxication) could result in an exaggerated sense of prospective confidence. This framework for quantifying prospective and retrospective cues to confidence can be applied to several modalities, including but not limited to driving [86], sports [103], gaming [104], and robotics and artificial limbs for patient rehabilitation [105].

## Conclusion

We demonstrated that the majority of our participants used prospective cues when determining their confidence in the outcome of a reach. Although all participants can use proprioception when prompted, as shown by our control motor-awareness task, only about a third of our participants additionally incorporated this retrospective cue when making a confidence judgement in the absence of visual feedback. The presence of multiple strategies across our participants adds support to the idea that dynamic metacognition is present throughout the action, and knowledge of the noise in various inputs allows for efficient weighting of the evidence.

## Supporting information

**S1 Fig. Simulations of the control experiment.** To test that a sufficient number of trials were collected the motor-awareness task, simulations were run for using various ground-truth parameter values. The low recovery error achieved with 300 trials was satisfactory. 100

iterations of 300 trials per simulated dataset. **A)** Estimates of motor error and ground-truth value. **B)** Estimates of proprioceptive error with a low sigma: very high accuracy. **C)** Estimates of proprioceptive error with a high sigma: less accurate but still close and unbiased.
(TIF)

**S2 Fig. Control motor awareness task true reach endpoints.** Endpoints (blue) shown around target location (black). For this task the target was in the same location on all trials.
(TIF)

**S3 Fig. Control motor awareness task reported endpoints.** Reported endpoints (red) rotated and shifted so that the origin is the reach endpoint (blue). The *y*-axis is thus the radial direction (along the reach) and the *x*-axis is the tangential direction, orthogonal to the reach. The target was in the same location on all trials. However, the reach endpoints did not always hit the target spot on. When comparing these scatterplots to those in Fig. 14 you can see the influence of the prior (target location) on the reported endpoints as they show the opposite bias of the reaches.
(TIF)

**S4 Fig. Scatter plot of reach endpoints and average confidence for all participants.** For the majority of our participants a Gaussian distribution was a good fit for the reach errors. Endpoint locations are rotated so that the *y*-axis is along the direction from start point to target and the *x*-axis is orthogonal to that direction. The red circle overlay shows the average confidence circle size in the center and ± SD for each participant as the transparent overlay.
(TIF)

**S5 Fig. Model recovery.** We performed a model recovery analysis to determine whether our experimental design would allow us to successfully recover the model that generated the data. We simulated data using 20 parameter sets. These 20 sets of $\{\sigma_m, \sigma_p, \sigma_s\}$ were paired with each of our three models of confidence judgments, resulting in a total of 60 simulated datasets. Each simulation (i.e., combination of parameter set and confidence model) consisted of 300 simulated trials for Task 1 and 900 simulated trials for Task 2 (300 confidence reports and 600 reaches alone). The parameter values for each simulated data set were sampled from log-normal distributions with statistics based on our expectations for the experiment given participants' performance (with mean and variance of $\sigma_m$: 20 mm and 25 mm$^2$, $\sigma_p$: 35 mm and 400 mm$^2$, $\sigma_s$: 7 mm and 25 mm$^2$). Each model was fit to each simulated dataset by maximum likelihood and the best-fitting model was determined by BIC. For 56 of the 60 data sets, the correct underlying model was selected. This robust recovery indicates that the number of experimental trials is sufficient to identify the model that generated the data. The numbers in each square are the number of simulations that were best fit by a given model.
(TIF)

**S6 Fig. Parameter recovery.** The best-fit parameters recovered by maximum likelihood for the ideal performance model under-predict the $\sigma_p$ parameter when there is a model mismatch and ground truth $\sigma_p$ is high (mean signed error in recovered parameter estimate as percentage of ground-truth value when $\sigma_p > 30$: 48%, SD: 16%) when the values were drawn from a log normal distribution with a mean of 40 and variance of 200 (S6A Fig). The degree of under-prediction was reduced when $\sigma_p$ was drawn from a log normal distribution with a mean of 20 and variance of 200, the error in the parameter estimate as a percentage of the ground-truth value when $\sigma_p < 30$: 16.7%, SD: 9.1%. The other two parameters were recovered well regardless of the $\sigma_p$ value, $\sigma_m$: 2.2%, SD:1.3%; $\sigma_s$: 15.7%, SD: 19.5%. Recovery of all three parameters was excellent for both the retrospective model ($\sigma_m$: 2.1%, SD:1.3%; $\sigma_p$: 3.6%, SD: 3.8%; $\sigma_s$: 5.4%, SD: 4.4%) and the prospective model ($\sigma_m$: 2.8%, SD:1.3%; $\sigma_p$: 6.5%, SD: 4.3%; $\sigma_s$: 3.1%, SD:

2.7%). Data generated using the ideal model (green), the retrospective model (blue) and the prospective model (red). Parameters are indicated by the shape of the data point: motor noise (star), proprioceptive noise (plus) and setting noise (square). **A)** Parameter recovery for the ideal model for data generated using the ideal model (green), the retrospective model (blue) and the prospective model (red). **B)** Parameter recovery for the retrospective model for data generated using the ideal model (green), the retrospective model (blue) and the prospective model (red). **C)** Parameter recovery for the prospective model model for data generated using the ideal model (green), the retrospective model (blue) and the prospective model (red).
(TIF)

**S7 Fig. Confidence vs Reach Error.** Correlations between Euclidian reach error distance and confidence circle size. Participants best fit by the Ideal performance model were more likely to have a significant correlation than those best fit by the Retrospective or Prospective models. Confidence circle size compared to reach error on each trial for all participants in Task 2. The green shaded area reflects judgements where the participant was able to earn points for enclosing their endpoint, and the red shaded area reflects judgements where the confidence circle was smaller than the distance from the endpoint to the target resulting in no points being awarded. *: Pearson's *r* significant at the .05 level.
(TIF)

**S8 Fig. Average confidence reports across all trials.** Average confidence is plotted +/- standard deviation for each participant compared to that participant's motor noise. We can see a positive relationship between motor noise and confidence across participants with a Pearson's *r* correlation of 0.5, significant at the .05 level.
(TIF)

**S9 Fig. Posterior predictive model comparisons.** To compare our models to the data we simulated sets of maximum expected gain confidence circle sizes with each of the three models utilizing the target locations and endpoints from each participant's data. The circle sizes simulated by the best-fitting model for each participant are the closest to the true circle sizes selected by that participant. Histograms showing selected circle size (yellow) for each participant. Data were simulated based on all three models using each participant's parameters and reach endpoints, the distributions of circle sizes selected by the models are juxtaposed over the data. The ideal model (green), retrospective model (blue) and prospective model (red) are all shown.
(TIF)

**S10 Fig. Trial-specific confidence rating comparison between models and data.** Scatter plots showing the circle sizes picked by each model compared to the actual circle sizes picked by the participant. The dashed line is the identity line. Participants 1–5 were best fit by the ideal model, and the circle sizes selected by that model are the closest match for those generated by the participant. The same can be seen for participants 7–16 as there is the greatest overlap between the prospective model circles and those chosen by the participant. Note that the small correlation in each cloud is due to the domination by real (for the data) and simulated (for the model) proprioceptive measurement noise.
(TIF)

## Acknowledgments

We would like to thank Eero Simoncelli for advice on model fitting and Zinong Li for assistance with data collection. This work utilized the NYU IT High-Performance Computing resources and services.

## Author Contributions

**Conceptualization:** Marissa E. Fassold, Shannon M. Locke, Michael S. Landy.

**Data curation:** Marissa E. Fassold.

**Formal analysis:** Marissa E. Fassold.

**Funding acquisition:** Michael S. Landy.

**Investigation:** Marissa E. Fassold.

**Methodology:** Marissa E. Fassold, Shannon M. Locke, Michael S. Landy.

**Validation:** Marissa E. Fassold.

**Visualization:** Marissa E. Fassold.

**Writing – original draft:** Marissa E. Fassold.

**Writing – review & editing:** Marissa E. Fassold, Shannon M. Locke, Michael S. Landy.

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
