## [Decision Letter · Decision Letter 0]

10 Feb 2023

Dear Mrs. Fassold,

Thank you very much for submitting your manuscript "Feeling lucky? Prospective and retrospective cues for sensorimotor confidence" for consideration at PLOS Computational Biology.

As with all papers reviewed by the journal, your manuscript was reviewed by members of the editorial board and by several independent reviewers. In light of the reviews (below this email), we would like to invite the resubmission of a significantly-revised version that takes into account the reviewers' comments.

Hope the constructive reviews can help strengthen the manuscript.

We cannot make any decision about publication until we have seen the revised manuscript and your response to the reviewers' comments. Your revised manuscript is also likely to be sent to reviewers for further evaluation.

Sincerely,

Ming Bo Cai

Academic Editor

PLOS Computational Biology

Daniele Marinazzo

Section Editor

PLOS Computational Biology

Hope the constructive reviews can help strengthen the manuscript.

Reviewer's Responses to Questions

**Comments to the Authors:**

Reviewer #1: Reviewer: Jonathan S. Tsay.

Summary: Fassfold et al asked what cues are used to estimate endpoint hand position and to estimate the confidence associated with hitting the target. They found that participants behave like an ideal observer when proprioceptive signals are more reliable, that is, they incorporate both retrospective (i.e., a belief that the hand is at the target, informed by participants’ movement history) and prospective cues (i.e., input from proprioceptive receptors on the current reach) in these estimates. However, when proprioceptive signals are less reliable, participants tend to rely only on retrospective cues. Together, these data shed light on the sensorimotor constraints underlying confidence judgments about hitting the target, taking us a step closer to understanding how our awareness of sensorimotor performance is determined.

Assessment: Fassfold et al tackled very important and timely questions. The methods and modeling are sound. The results are clear. I have a few suggestions below that can hopefully strengthen the manuscript:

1) Confidence in hitting the target vs confidence in positional estimates. I understand that sensorimotor confidence refers to the confidence in successfully hitting the target, hence, that is why the circle is centered on the actual target location. It makes complete sense why prior information based on a history of motor errors (i.e., motor noise) is used here to make a judgment. I have no qualms about the underlying (and elegant) Bayesian principles outlined in the manuscript.

However, could the current task bias the results toward relying on prior information? That is, the participants are asked to return to the start position before making a confidence judgment, as opposed to holding the actual hand position in its original location when making a confidence judgment (e.g., (Ruttle et al., 2021; Tsay et al., 2020), but note that these are not confidence judgments but positional judgements). The latter simulates a real-life scenario in which one reaches forward to grab an object, misses rightward, detects a rightward error presumably using prospective proprioceptive cues in an online manner, and makes a corrective movement leftward. Together, it seems like moving the hand back to the start position may introduce potential forgetting or interference (i.e., proprioceptive cues from other locations), which may decrease the relative weight of proprioceptive cues on the confidence judgments. This would predict that the weight of proprioceptive cues may be used more if the task were designed differently, and potentially introduce alternative models (ideal observer + forgetting rate dependent on time; ideal observer + interference) for future tests.

In addition, if the task were to estimate the confidence around the positional estimate of the hand after a movement (i.e., setting the confidence ring around the hand, rather than the target), would the authors expect participants to give greater weight to prospective proprioceptive cues rather than retrospective cues? I understand that this question may be outside the scope of the current study, but seems like a possible caveat to discuss since it may speak to how a little tweaks in the question/task could potentially bias use of one cue over another.

2) What signals inform the development of participants’ prior belief that their hand is positioned at the target? The authors suggest that the prior belief is informed by/related to motor noise. However, could the authors provide a bit more intuition about the sensory origin around this belief? Is it that, during the control task, participants build up an internal representation about the distribution of the visual cursor feedback signaling their actual hand position, and as such, the prior belief is informed by vision? Therefore, the ideal observer model is thus an integration between a memory of the visual distribution against a nosier proprioceptive distribution? Or, is the prior belief constructed by relying on a history of prospective proprioceptive cues (without vision) integrating with a flat prior on a trial-by-trial basis, and that this prior is constructed slowly by the distribution of estimated endpoints? That is, prior motor noise may be dependent/interact with proprioceptive noise. Or do the authors think that confidence judgments access ‘motor noise’, a signal that is not tied to any specific sensory origin?

3) Reliability of the data. The authors mention a control motor awareness task to estimate motor noise and proprioceptive noise. Would the authors be open to representing these data in the supplemental figure, especially a potential figure showing the reliability of motor/proprioceptive noise + fitting parameters on half of the data, correlate with fits on the other half? This seems like an important figure that would lend greater credence to the quality of the data in their main task.

4) Validity of the confidence judgment. I think the confidence judgments are a cool continuous measure that moves us away from binary estimates of confidence (hit vs miss; side note: I did really like the following paper by the group that used binary judgments: (Gaffin-Cahn et al., 2019)). That being said, out of the 16 participants, only 8 show a significant correlation between confidence judgments and reach errors (Figure 17). Perhaps to complement this weak association, the authors could contrast this with a between-participants correlation of motor noise vs confidence judgments, to show that this confidence measure indeed tracks a behavioral measure.

The largely invariant confidence judgments within-participant may indeed reflect insensitivity to proprioception (for most individuals). But alternatively, it may simply reflect a strategy (risk aversion in setting a ring size that incorporates most motor noise), as the authors outlined in the Discussion. In addition, it may also reflect ‘laziness’ in their adjustments (setting the ring pretty much at a similar location), masquerading sensitivity to proprioceptive cues – this may reflect a response bias. To this end, it was not clear in the manuscript whether the authors jittered the initial size of the confidence ring before the participant’s judgment to counterbalance out/discourage possible response biases. If so, it would be helpful to detail the starting position of the ring in the Methods section.

Awesome work. I enjoyed reading the manuscript, thoroughly.

References

Gaffin-Cahn, E., Hudson, T. E., & Landy, M. S. (2019). Did I do that? Detecting a perturbation to visual feedback in a reaching task. Journal of Vision, 19(1), 5.

Ruttle, J. E., Hart, B. M. ’t, & Henriques, D. Y. P. (2021). Implicit motor learning within three trials. Scientific Reports, 11(1), 1627.

Tsay, J. S., Parvin, D. E., & Ivry, R. B. (2020). Continuous reports of sensed hand position during sensorimotor adaptation. Journal of Neurophysiology, 124(4), 1122–1130.

Reviewer #2: This work addresses the interesting question what determines sensory-motor confidence. The authors hypothesize that people integrate multiple cues and design a task that affords continuous confidence measures and incentivized both accurate performance and confidence judgments. They extend a Bayesian model of sensory-motor behavior to confidence judgments in this task. They fit competing models jointly to behavior on this task, as well as a control task, allowing them to constrain and estimate parameters that are otherwise not observable in the main task. They compare three models: A optimal Bayesian observer that integrates prospective information based on past observations, a retrospective model that only uses current, proprioceptive evidence, and a prospective model, that only uses past experience. The authors find strong individual differences in which model captures behavior better, and that people with more reliable proprioception are more likely to integrate that information into their confidence judgments.

I found this study interesting, insightful, the experiment clever, the analyses rigorous, and the manuscript generally very clear. I have a minor comment and a question, but overall found this manuscript a pleasure to read.

One thing I found confusing is that in the figures for the different models the confidence circle was always the same size. That size should vary depending on where the estimated end-point is, thought, right? I think changing that would help clarify the different predictions these models make that allow them to be tested against each other. Perhaps visualizing that in one shared figure would also help tune people's intuitions.

Conceptually, I have a question about whether this procedure measures confidence per se versus error monitoring - that is without an uncertainty estimate associated with it, as in perfect endpoint estimation - and whether the authors think of these functions as distinct versus integrated. I understand that for continuous measures the conceptual space is still very muddy, so I don't expect the authors to necessarily address that in the paper - I am curious, though.

Reviewer #3: Review of "Feeling lucky? Prospective and retrospective cues for sensorimotor confidence"

This is an interesting and well-written paper combining experiment and modelling which studies the role of prospective and retrospective cues in a pointing task.

The main novelty here is the usage of a "interval-like" report (a circle) with a scoring system which would encourage subjects to report their uncertainty/confidence about their distance to the target on a continuous scale. Similar methods have been used extensively in perceptual studies of uncertainty/confidence (e.g., Honig, Ma & Fougnie, PNAS 2020), but I am not aware of this paradigm used in a sensorimotor study.

I think the paper is a very nice contribution to the sensorimotor modelling literature, with a few comments below.

Major comments:

1. Model fit visualization:

Everything is well-executed and well-presented in terms of experimental design, computational modelling, model fitting, model comparson (with minor caveats), model recovery. However, the current submission does not show much about the model fits / posterior-predictive checks. In some sense, model comparison is meaningful only when the model set contains the true model (or, realistically, a close-enough model), otherwise it is just a model-fitting exercise (an exercise I am quite fond of, but still). In practice, the reader needs to be convinced that the best-fitting models are effectively capturing some key properties of the data, and this is done by showing the model fits to various aspects of the data.

In this paper, as far as I can see, Fig 12 (and Fig 18 in the Appendix) are the only "fits" that we are shown. With a bit of a skeptical note, we could say that the distribution of "confidence circle size" in Fig 18 is for most subjects somewhere in the middle of the range, and the best-fitting model for each subject ends up simply being whichever model overlaps most with this distribution (sometimes it is one model, sometimes it is another model). Can the authors show some other summary statistics / observable of the data to indeed provide a stronger feel for how the models fit the data?

For example, instead of a distribution/histogram plot, the authors could show a scatter plot of true circle size vs. predicted circle size (for different models), so that at least we can see that there is a correlation between model predictions and actual responses on a trial-by-trial basis (which we cannot see from Fig 12/18), which would be a stronger evidence that the model is capturing a real process. Maybe there are also other plots that the authors can present to strengthen the model fit presentation.

2. Alternative models?

As a follow-up to point 1: given the relatively little information about the model fits themselves - and part of the Discussion -, I am under the impression that no model fits the data particularly well here (but happy to be corrected by additional plots or analyses).

In particular, the reported model-switching between "ideal" and "prospective" model for different values of the proprioceptive uncertainty parameter is especially suggestive that there may be some other mechanism at play, which is not captured by any current model. The "switch" of the model, as mentioned previously, may just be accidental (i.e., the winning model is whichever happens to overlap with the circle distribution, which is dictated by the true underlying model).

Just to be clear, some bad fits (and a lack of a "true model") are common in pychophysics and we all agree it's fine as long as some important aspects of the data are captured by the presented models. However, I think in this case it's worth for the authors to give a thought about alternative models, especially here since it seems that there is indeed some underlying pattern. I am not going to give a specific advice since the goal is not to fit "that reviewer's favourite hypothesis", but just to think whether there might be other models to be considered. (There are plenty of ideas to be taken from the perceptual and decision-making domain.)

Minor comments:

- "For our model fits of behavioral data, the smallest BIC difference between the winning model and next-best model was 12.38, which is already strong evidence as guidelines suggest a score over 10 to be vigorously compelling (Raftery, 1995)."

I used to write this stuff as well (everybody does), but just a remark that this may be somewhat misleading. The statement above is correct when the true model is in the model set, but since this is never the case, its *interpretation* as "vigorously compelling" is debatable.

- First, differences of AIC/BIC in the hundreds are very common for a typical psychophysical experiment with hundreds of trials, but that's just a consequence of the concentration of probability. The fact that we are much more certain that a model is better than another does not mean that the winning model is anywhere close to the true model (I know the authors are not stating that, but the less experienced reader might be confused).

- Second, here we are "very certain" (i.e., large BIC difference) that some subjects are following the "ideal" model, and we are also "very certain" that some other subjects are following the "prospective" model, but this is also exactly what one would expect under model misspecification (i.e., lack of a true model). One possibility is that the true model is in some other parts of model space, and what happens here is that whichever model ends up slightly closer (for whatever reason) to the true model ends up winning by seemingly a large amount, but this is effectively just "noise" in model space.

**Have the authors made all data and (if applicable) computational code underlying the findings in their manuscript fully available?**

Reviewer #1: Yes

Reviewer #2: **No: **In the availability statement is says that everything will be available, but I couldn't find a link. It's possible that there is one in another place in the manuscript, in which case that just needs to be added in the relevant section.

Reviewer #3: Yes

PLOS authors have the option to publish the peer review history of their article (what does this mean?). If published, this will include your full peer review and any attached files.

Reviewer #1: **Yes: **Jonathan S. Tsay

Reviewer #2: No

Reviewer #3: **Yes: **Luigi Acerbi
---

## [Decision Letter · Decision Letter 1]

9 May 2023

Dear Fassold,

Thank you very much for submitting your manuscript "Feeling lucky? Prospective and retrospective cues for sensorimotor confidence" for consideration at PLOS Computational Biology. As with all papers reviewed by the journal, your manuscript was reviewed by members of the editorial board and by several independent reviewers. The reviewers appreciated the attention to an important topic. Based on the reviews, we are likely to accept this manuscript for publication, providing that you modify the manuscript according to the review recommendations.

I agree with Reviewer 3's suggestion to provide a clearer evaluation of model fit by comparing some curves of summary statistics derived from subjects' data and what the model predicts for the same metrics (e.g., using the parameters fitted from the data to simulate behavior and calculate the same summary statistics) to show that the model can indeed capture patterns in the real data. Even if the model's prediction has some deviation from the real data's curve, there is nothing to be shy about it as this is common. But pointing out the limitation (if there is) is helpful for readers to have a good evaluation of the model and potentially develop new works inspired by your paper.

Sincerely,

Ming Bo Cai

Academic Editor

PLOS Computational Biology

Daniele Marinazzo

Section Editor

PLOS Computational Biology

I agree with Reviewer 3's suggestion to provide a clearer evaluation of model fit by comparing some curves of summary statistics derived from subjects' data and what the model predicts for the same metrics (e.g., using the parameters fitted from the data to simulate behavior and calculate the same summary statistics) to show that the model can indeed capture patterns in the real data. Even if the model's prediction has some deviation from the real data's curve, there is nothing to be shy about it as this is common. But pointing out the limitation (if there is) is helpful for readers to have a good evaluation of the model and potentially develop new works inspired by your paper.

Reviewer's Responses to Questions

**Comments to the Authors:**

Reviewer #1: Thank you for addressing all my comments and suggestions, and for your contributions in this paper.

JT

Reviewer #2: The authors have answered my questions and addressed my concerns. I believe this manuscript will make a great contribution to the literature.

Reviewer #3: Thanks to the authors for their careful revision and for addressing most of my concerns with changes in the paper.

However, my concern about the quality of the model fits is still there. The additional scatter plots (Fig S9) do not quite address my concerns. In Fig S9, there seems to be little correlation between the data and the model; the authors acknowledge that in the caption, but it doesn't change the fact that these model fits don't do much to showcase goodness of fit.

If adding simulated noise obfuscates whether the models are good or not, as I suggested in my previous review, the authors could come up with a relevant summary statistic (averaged over some trials or relevant conditions) that can be plotted over the model prediction (with error bars); a common practice to show an approximate match between data and model.

Overall, to better restate my main point: my only concern about this study is that the final analysis misses a critical assessment of whether the models truly capture features of the data. The model evaluation step is an essential part of the modelling workflow (e.g., see Gelman et al., 2020, "Bayesian workflow"). I encourage the authors to expand the model evaluation in this paper, e.g. producing some plots and analyses that support the validity of their models (some of which should be in the main text, and not relegated to the Appendix).

For example, the authors could indeed show that their models capture relevant features of the data for multiple subjects (maybe some of the existing plots in the Appendix already do that; in which case it's enough to point the reader and highlight the relevant results). Conversely, the authors might come to acknowledge that the model fits are overall not great (as it is often the case in psychophysics!); but nonetheless they could argue that their study is still informative for this and that reason, which I would be also fine with.

To conclude, I think this is overall a good study, and I very much like the premise, the experiments and the modelling setup; I just feel the results (model fits, how they quantitatively and qualitatively relate to the data) and in particular the model evaluation part could be expanded; and this would strengthen the paper by convincing the (nitpicky) reader of the validity of the results.

**Have the authors made all data and (if applicable) computational code underlying the findings in their manuscript fully available?**

Reviewer #1: Yes

Reviewer #2: Yes

Reviewer #3: Yes

PLOS authors have the option to publish the peer review history of their article (what does this mean?). If published, this will include your full peer review and any attached files.

Reviewer #1: **Yes: **Jonathan Sanching Tsay

Reviewer #2: No

Reviewer #3: **Yes: **Luigi Acerbi

Figure Files:

Data Requirements:

Reproducibility:

References:

---

## [Decision Letter · Decision Letter 2]

8 Jun 2023

Dear Fassold,

We are pleased to inform you that your manuscript 'Feeling lucky? Prospective and retrospective cues for sensorimotor confidence' has been provisionally accepted for publication in PLOS Computational Biology.

Best regards,

Ming Bo Cai

Academic Editor

PLOS Computational Biology

Daniele Marinazzo

Section Editor

PLOS Computational Biology

Reviewer's Responses to Questions

**Comments to the Authors:**

Reviewer #3: Thanks to the authors for the additional analyses. I am happy with the new figure comparing data and model fits on several summary statistics.

**Have the authors made all data and (if applicable) computational code underlying the findings in their manuscript fully available?**

Reviewer #3: Yes

PLOS authors have the option to publish the peer review history of their article (what does this mean?). If published, this will include your full peer review and any attached files.

Reviewer #3: **Yes: **Luigi Acerbi

---

## [Editor Report · Acceptance letter]

21 Jun 2023

PCOMPBIOL-D-22-01668R2 

Feeling lucky? Prospective and retrospective cues for sensorimotor confidence

Dear Dr Fassold,

I am pleased to inform you that your manuscript has been formally accepted for publication in PLOS Computational Biology. Your manuscript is now with our production department and you will be notified of the publication date in due course.

With kind regards,

Timea Kemeri-Szekernyes
